# The Nesprin-1/-2 ortholog ANC-1 regulates organelle positioning in *C. elegans* independently from its KASH or actin-binding domains

Hongyan Hao, Shilpi Kalra, Laura E Jameson, Leslie A Guerrero, Natalie E Cain, Jessica Bolivar, Daniel A Starr*

Department of Molecular and Cellular Biology, University of California, Davis, Davis, United States

**Abstract** KASH proteins in the outer nuclear membrane comprise the cytoplasmic half of linker of nucleoskeleton and cytoskeleton (LINC) complexes that connect nuclei to the cytoskeleton. *Caenorhabditis elegans* ANC-1, an ortholog of Nesprin-1/2, contains actin-binding and KASH domains at opposite ends of a long spectrin-like region. Deletion of either the KASH or calponin homology (CH) domains does not completely disrupt nuclear positioning, suggesting neither KASH nor CH domains are essential. Deletions in the spectrin-like region of ANC-1 led to significant defects, but only recapitulated the null phenotype in combination with mutations in the transmembrane (TM) span. In *anc-1* mutants, the endoplasmic reticulum ER, mitochondria, and lipid droplets were unanchored, moving throughout the cytoplasm. The data presented here support a cytoplasmic integrity model where ANC-1 localizes to the ER membrane and extends into the cytoplasm to position nuclei, ER, mitochondria, and other organelles in place.

*For correspondence:
dastarr@ucdavis.edu

Competing interests: The authors declare that no competing interests exist.

## Introduction

Cellular organization is an essential process. Organelles are interconnected and mostly constrained to specific subcellular locations when they are not actively transported longer distances by cytoskeletal motor proteins (*van Bergeijk et al., 2016*). For example, nuclear positioning is essential for a wide variety of cellular and developmental processes, including fertilization, cell division, cell polarization, gametogenesis, central-nervous system development, and skeletal muscle function (*Bone and Starr, 2016*; *Gundersen and Worman, 2013*). Defects in nuclear positioning are associated with multiple neuromuscular diseases (*Calvi and Burke, 2015*; *Folker and Baylies, 2013*; *Gundersen and Worman, 2013*). Furthermore, the Golgi apparatus and centrosomes are often found next to the nucleus, while the ER and mitochondria are usually spread throughout the cell (*van Bergeijk et al., 2016*). The cellular tensegrity model posits that organelles are physically coupled to the cytoskeleton, plasma membrane, and extracellular matrix so that the cell acts as a single mechanical unit (*Jaalouk and Lammerding, 2009*; *Wang et al., 2009*). The mechanisms that maintain cellular tensegrity and how it relates to organelle positioning are poorly understood, especially in vivo.

Nuclei are connected to the rest of the cell by LINC (linker of nucleoskeleton and cytoskeleton) complexes. SUN (Sad-1/UNC-84) proteins integral to the inner nuclear membrane and KASH (<u>K</u>larsicht, <u>ANC-1</u>, and <u>SYNE</u> <u>h</u>omology) proteins that span the outer nuclear membrane interact with each other in the perinuclear space to form LINC complexes. The cytoplasmic domains of KASH proteins interact with various components of the cytoskeleton (*Luxton and Starr, 2014*), while the nucleoplasmic domains of SUN proteins interact with lamins. Thus, LINC complexes bridge the

nuclear envelope and mechanically couple the nucleoskeleton to the cytoskeleton (*Chang et al., 2015*; *Lee and Burke, 2018*; *Starr and Fridolfsson, 2010*). LINC complex inhibition reduces the stiffness and increases the deformability of the entire cytoplasm in mammalian tissue culture cells, even far from the nuclear envelope and beyond the predicted reach of LINC complexes (*Gill et al., 2019*; *Stewart-Hutchinson et al., 2008*). Whether LINC complexes maintain the mechanical properties of the cytoplasm in vivo is relatively unexplored.

Here, we investigate the role of LINC complexes with giant KASH proteins in organelle positioning in *Caenorhabditis elegans*. Most of the hypodermis of an adult *C. elegans* consists of a giant syncytium, the hyp7, containing 139 evenly-spaced nuclei that are anchored in place (*Altun and Hall, 2009*). Furthermore, *C. elegans* have invariant developmental lineages, are optically clear, and are easily genetically manipulated, making the hyp7 an ideal in vivo model to study organelle positioning. A LINC complex made of the KASH protein ANC-1 and the SUN protein UNC-84 is responsible for nuclear anchorage in *C. elegans* (*Starr, 2019*). UNC-84 is a canonical SUN protein that is orthologous to mammalian SUN1 and SUN2 and is the only known SUN protein in postembryonic somatic tissues in *C. elegans* with a nucleoplasmic domain that interacts directly with the lamin protein LMN-1 (*Bone et al., 2014*). ANC-1 is an exceptionally large protein of up to 8545 residues with two tandem calponin homology (CH) domains at its N terminus and a KASH domain at its C terminus (*Starr and Han, 2002*). ANC-1 orthologs *Drosophila* MSP-300 and mammalian Nesprin-1 Giant (G) and -2G have similar domain arrangements (*Starr and Fridolfsson, 2010*). Unlike MSP-300, Nesprin-1G, and -2G, which each contains greater than 50 spectrin-like repeats, ANC-1 consists of six tandem repeats (RPs) of 903 residues that are almost 100% conserved with each other at the nucleotide level (*Liem, 2016*; *Rajgor and Shanahan, 2013*; *Starr and Han, 2002*; *Zhang et al., 2001*). While spectrin-like repeats have not been identified in the ANC-1 RPs, most of ANC-1 is predicted to be highly helical, like spectrin (*Starr and Han, 2002*). The CH domains of ANC-1 interact with actin filaments in vitro and co-localize with actin structures in vivo, while the KASH domain of ANC-1 requires UNC-84 for its localization to the outer nuclear membrane (*Starr and Han, 2002*). UNC-84 is thought to interact with lamins to connect LINC to the nucleoskeleton while ANC-1 extends away from the outer nuclear membrane into the cytoplasm to tether nuclei to actin filaments (*Starr and Han, 2002*).

Evidence from multiple systems suggests that giant KASH orthologs might not solely function as nuclear tethers. We have observed that hyp7 syncytia in *anc-1* null animals display a stronger nuclear positioning defect than *unc-84* null animals (*Cain et al., 2018*; *Jahed et al., 2019*) and mitochondria are unanchored in *anc-1*, but not in *unc-84* mutants (*Starr and Han, 2002*). These results suggest that ANC-1 has LINC complex-independent roles for anchoring nuclei and mitochondria. Likewise, mitochondria and the ER are mispositioned in *Drosophila msp-300* mutant muscles (*Elhanany-Tamir et al., 2012*). Finally, it remains to be determined if the CH domains of ANC-1 are necessary for nuclear anchorage. Mouse Nesprin-1 and -2 have isoforms lacking CH domains (*Duong et al., 2014*; *Holt et al., 2016*) and Nesprin-1 CH domains are dispensable for nuclear positioning during mouse skeletal muscle development (*Stroud et al., 2017*).

To address these ambiguities, we use ANC-1 to examine how giant KASH proteins position nuclei and other organelles. We find that for nuclear anchorage, the ANC-1 KASH domain plays a relatively minor role, and the CH domains are dispensable. Rather, multiple large cytoplasmic domains and the C-terminal transmembrane (TM) span of ANC-1 are required for nuclear anchorage. Moreover, in *anc-1* null mutants, the entire cytoplasm is disorganized, and the ER is unanchored and moves freely throughout the cytoplasm. Together, our results support a model in which ANC-1 associates with ER membranes to regulate the mechanical properties of the cytoplasm, thereby anchoring nuclei, mitochondria, ER, and other organelles.

## Results

### ANC-1 promotes proper nuclear anchorage through a LINC complex-independent mechanism

Loss-of-function mutations in *anc-1* disrupt the even spacing of hyp7 syncytial nuclei (*Cain et al., 2018*; *Starr and Han, 2002*). We use the number of nuclei in contact with each other as a metric for hyp7 nuclear anchorage defects (*Cain et al., 2018*; *Fridolfsson et al., 2018*). In wild type

(WT) animals, very few hyp7 nuclei were touching. In contrast, over 50% of hyp7 nuclei were clustered with at least one other nucleus in *anc-1(e1873)* null mutants. Significantly fewer hyp7 nuclei were unanchored in *unc-84(n369)* null mutants (*Figure 1C*). This trend was also observed in adult syncytial seam cells (*Figure 1—figure supplement 1*). Throughout this manuscript, we call nuclear anchorage defects that are statistically similar to *anc-1* null mutants as severe, defects statistically similar to *unc-84* null mutants but still significantly worse than wild type as mild, and defects statistically between *anc-1* and *unc-84* null mutants as intermediate.

The greater severity of the nuclear anchorage defect in *anc-1* null mutants compared to *unc-84* suggests that ANC-1 plays additional roles in nuclear positioning independently of its SUN partner UNC-84. We used CRISPR/Cas9 gene editing to delete the luminal peptides of the ANC-1 KASH domain (*Figure 1D*). We predicted the *anc-1(ΔKASH)* mutants would abrogate the interaction between ANC-1 and UNC-84 and phenocopy *unc-84(null)* animals. Two independent *anc-1(ΔKASH)* mutants exhibited mild nuclear anchorage defects similar to those observed in *unc-84(null)* mutants (*Figure 1B–C*). Together, these results suggest that the SUN/KASH interaction only partially contributes to nuclear anchorage, implicating the large cytoplasmic domain of ANC-1 as the major player in nuclear positioning.

## The ANC-1 N-terminal CH domains are not required for hyp7 nuclear positioning

We next deleted the CH domains at the N terminus of the largest isoforms of ANC-1, which are predicted to interact with actin, and replaced them with GFP using CRISPR/Cas9 gene editing. Hyp7 nuclei did not cluster in *anc-1(ΔCH)* mutants (*Figure 2B*). Thus, the CH domains of ANC-1 are not required for hyp7 nuclear anchorage.

Nesprin-1 and -2 have multiple splice isoforms, many of which are missing the CH domains (*Rajor et al., 2012*; *Stroud et al., 2017*). We hypothesized a shorter *anc-1* isoform lacking the CH domains would be sufficient for nuclear anchorage. RNAseq and expressed sequence tag data published on WormBase (*Harris et al., 2020*) suggest that *anc-1* has at least three isoforms (*Figure 2A*). We tested whether a shorter isoform lacking CH domains, *anc-1b*, is sufficient for hyp7 nuclear anchorage. RNAi constructs targeting the 5' exons specific to the *anc-1a/c* long isoforms did not cause nuclear anchorage defects (*Figure 2A–B*). However, RNAi targeting a repetitive region in all three predicted isoforms caused severe nuclear anchorage defects (*Figure 2A–B*). We also analyzed four nonsense mutations. Alleles that are predicted to disrupt the longer *anc-1a/c* isoforms but not the shorter *anc-1b* isoform, *anc-1(W427*)*, and *anc-1(W621*)* were normal for nuclear anchorage. In contrast, both *anc-1(Q1603*)* and *anc-1(Q2878*)* alleles, which are predicted to add premature stop codons to all three predicted isoforms, led to severe nuclear anchorage defects (*Figure 2A–B*). These results suggest that the shorter *anc-1b* isoform lacking the CH domains is sufficient for nuclear anchorage.

We next tested whether *anc-1b* is expressed in hyp7. First, 5' RACE (Rapid amplification of cDNA ends) was used to identify the start of the *anc-1b* predicted transcript (*Figure 2C*). The RACE product contained an SL1 sequence at its 5' end, suggesting this represents the end of a bona fide transcript (*Figure 2D*). To test if the *anc-1b* isoform is expressed in hyp7, we fused the *anc-1b* promoter and ATG to an *nls::gfp::lacZ* reporter and expressed it in transgenic animals. The *anc-1b* promoter drove GFP expression in hyp7 (*Figure 2E*, yellow arrows). Taken together, these results suggest the conserved CH domains are not necessary for hyp7 nuclear anchorage and the *anc-1b* isoform expressed in the hypodermis plays a major role in hyp7 nuclear anchorage.

## Spectrin-like domains of ANC-1b are required for nuclear anchorage

We predicted that the six tandem repeats (RPs) of ANC-1 function analogously to the spectrin-like domains of Nesprin-1 and -2. We modeled the structure of pieces of the ANC-1 RPs using the protein-folding prediction software QUARK (*Xu and Zhang, 2012*; *Xu and Zhang, 2013*) and found that they are predicted to form helical bundles remarkably similar to the structure of spectrin (*Figure 3A,D*; *Grum et al., 1999*), suggesting the tandem repeats are analogous to spectrin-like domains.

We next tested the necessity of the ANC-1 spectrin-like repeats and the neighboring cytoplasmic domains for nuclear anchorage by making in-frame deletions of portions of *anc-1* using CRISPR/

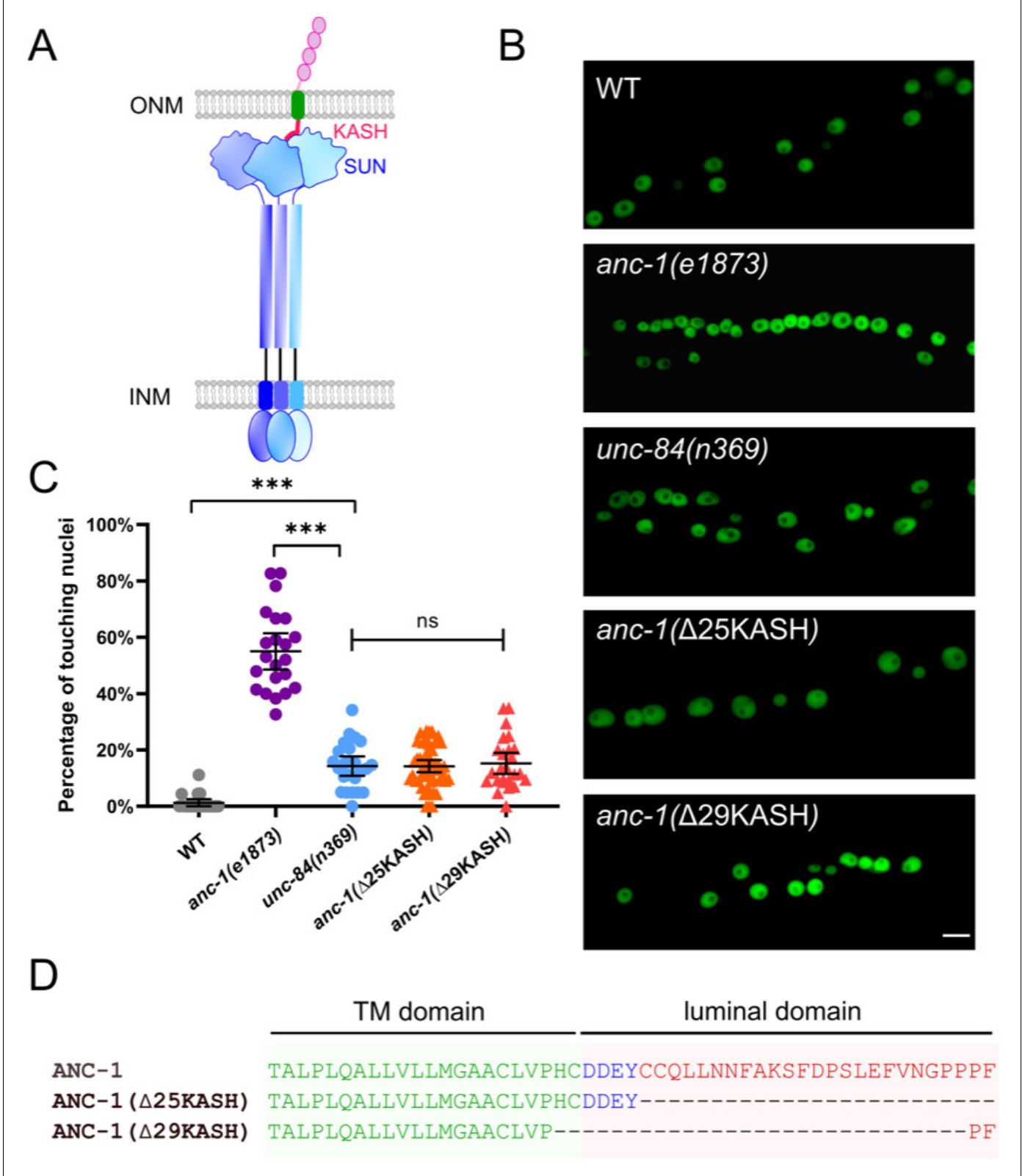

**Figure 1.** ANC-1 has a linker of nucleoskeleton and cytoskeleton (LINC) complex-independent role in anchoring nuclei. (A) Model of the LINC complex. Trimers of the SUN protein UNC-84 (purple and blue) and the KASH protein ANC-1 (red, green, and pink, only one of the trimers is shown) form the LINC complex, which spans the outer nuclear membrane (ONM) and inner nuclear membrane (INM). (B) Lateral views of adult *C. elegans* expressing hypodermal nuclear GFP in wild type (WT) or indicated mutants. Scale bar, 10 μm. (C) Quantification of nuclear anchorage defects. Each point
*Figure 1 continued on next page*

Figure 1 continued

represents the percentage of touching nuclei on one side of a young adult animal. Means with 95% CI error bars are shown. ANOVA and Tukey's multiple comparisons tests were used for statistical analysis; ns means not significant, p>0.05; ***p≤0.001. n ≥ 20 for each strain. (D) Sequences of the transmembrane (TM) domain and the luminal domain of ANC-1 showing the deletions analyzed.

The online version of this article includes the following figure supplement(s) for figure 1:

**Figure supplement 1.** Nuclear anchorage defects in seam cell syncytia.

Cas9-mediated gene editing. The N-terminal fragment 1 (F1) contains 1969 residues of ANC-1b from the start codon to the start of the RPs and fragment 2 (F2) contains the 277 residues between the RPs and the C-terminal transmembrane span (*Figure 3A*). The deletion of the F1 domain or all six RPs caused intermediate nuclear anchorage defects (*Figure 3B–C*). ANC-1b with only one of the normal six repeats had an intermediate nuclear anchorage defect that was significantly greater than in wild type but not as severe as the *anc-1(Δ6RPS)* mutant (*Figure 3C*). In contrast, the *anc-1(ΔF2)* mutant had no nuclear anchorage defect (*Figure 3C*). Since the *anc-1(Δ6RPS)* and *anc-1(ΔF1)* defects were less severe than *anc-1* null alleles, we made double mutants with *unc-84(n369)* to see if mutations in the cytoplasmic portions of ANC-1 were synergistic with mutations in the KASH domain. Both *anc-1(ΔF1); unc-84(n369)* and *anc-1(Δ6RPS); unc-84(n369)* double mutants significantly enhanced the nuclear anchorage defects of the single mutations (*Figure 3C*). However, the hyp7 nuclear anchorage defects in *anc-1(ΔF1); unc-84(n369)* and *anc-1(Δ6RPS); unc-84(n369)* double mutants were still less severe than *anc-1(e1873)* null mutants, suggesting that multiple parts of ANC-1b mediate proper hyp7 nuclear positioning (*Figure 3C*). Together, these results indicate that (1) F1 and the RPs play roles in nuclear anchorage, (2) multiple repeats are necessary for normal function, and (3) the F2 region is dispensable for hyp7 nuclear positioning.

## The ER is unanchored in *anc-1* mutants

In addition to nuclear positioning, ANC-1 functions in mitochondria distribution and morphology in the hypodermis and muscle cells (*Hedgecock and Thomson, 1982*; *Starr and Han, 2002*). We therefore asked if ANC-1 also anchors other organelles. We characterized the ER in live hyp7 syncytia of *anc-1* and *unc-84* mutants using a single-copy GFP::KDEL marker (*pwSi83*; gift of Barth Grant). In wild type, the ER formed a network evenly distributed throughout hyp7 (*Figure 4A*). We used blind scoring to classify single images of each animal's ER as normal (evenly distributed in what appeared to be sheets), mild defects (lots of sheet-like structures, but occasionally not uniformly spread throughout the syncytium), strong defects (considerable mispositioning and clustering of ER, but still in large units), or severe defects (complete mispositioning and extensive fragmenting of the ER) (*Figure 4* and *Figure 4—figure supplement 1*). About a third of wild type adults had normally distributed ER, while the rest had mild ER positioning defects, perhaps due to the pressure on the worm from the coverslip (*Figure 4A–B*). However, ER networks in *anc-1* null mutants were severely disrupted and often fragmented (*Figure 4A–B*). We also observed the dynamics of the ER in live animals. In wild type animals, the hypodermal ER was anchored as an interconnected network and exhibited limited motion while the animal crawled (*Figure 4C,E–F*, *Video 1*). However, in *anc-1* null mutants, ER fragments drifted apart and often formed large aggregates, suggesting that the anchorage of the ER network was disrupted (*Figure 4D–F*, *Video 2*). To quantify this phenotype, we measured the change in distance between distinct points over time. In wild type, the average distance change between parts of the ER was less than 1 μm per second, whereas in *anc-1* mutants, the average change in distance more than doubled (*Figure 4E*), suggesting that the ER had lost its overall interconnectivity and that fragments were unanchored from the rest of the ER.

We next examined whether UNC-84 is required for ER positioning. Still images of *unc-84(n369)* null mutants scored blindly were similar to wild type ER (*Figure 4A–B*). However, in some videos of *unc-84(n369)* mutants, there were slight changes in the organization of the ER over time (*Video 3*), suggesting that *unc-84* null mutants had a minor ER positioning defect. Most *anc-1(Δ29KASH)* mutants had mild defects in ER positioning, and only about a quarter had more severe defects, significantly less than *anc-1(e1873)* null mutants. In contrast, more than 80% of *anc-1(Δ6RPS)* animals had strong or severe ER positioning defects, similar to *anc-1* null mutants (*Figure 4A–B*). These

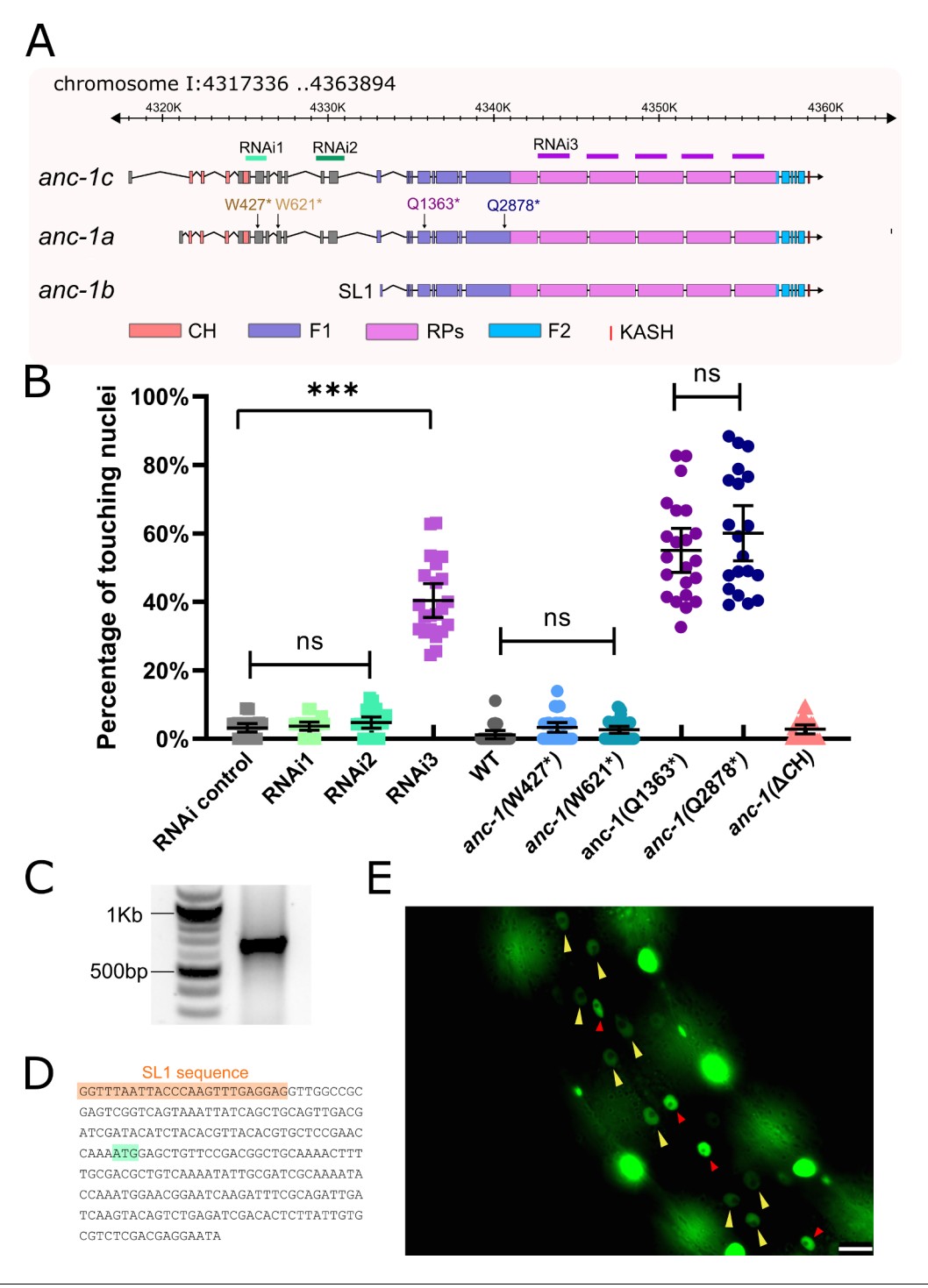

**Figure 2.** *anc-1b* is the major isoform in hyp7 nuclear anchorage. (**A**) Schematic gene structure of *anc-1a*, *b,* and *c* isoforms (modified from the J-Brower in Wormbase). Domains are color-coded. CH = calponin homology; F1 = fragment one from the ATG of ANC-1b to the beginning of the repeats; RPs = the six exact repeats of about 900 residues each; F2 = fragment two from the end of the repeats to the transmembrane (TM) span; WT = wild type; and KASH = Klarsicht, ANC-1, Syne homology, in this case referring to the residues in the lumen of the nuclear envelope. The target regions of RNAi constructs are labeled. Premature stop mutations are indicated using the numbering the *anc-1a* isoform. (**B**) Quantification of nuclear anchorage defects in *anc-1* mutant and RNAi animals. Means with 95% CI are shown in the graph. ANOVA and Tukey's multiple comparisons tests were used for statistical analysis. ns, not significant, p>0.05; ***p≤0.001. n ≥ 20 for each strain. (**C**) An agarose gel showing

*Figure 2 continued on next page*

*Figure 2 continued*

the 5'-RACE products on the right lane. (D) Partial sequence of the 5'-RACE product. An SL1 sequence (orange) adjacent to the 5' end of the *anc-1b* transcript was identified. The predicted start codon is in light green. (E) Lateral view of a worm showing the expression of nls::GFP driven by *anc-1b* promoter. Yellow arrows mark hyp7 nuclei; the red arrows mark seam cell nuclei; and the bright, unmarked nuclei are in muscle cells. Scale bar, 10 μm.

results indicated that ANC-1 is essential for ER positioning through mostly LINC complex-independent mechanisms.

## ANC-1 localizes to ER membranes

Since ANC-1 functions, in part, independently of LINC complexes at the nuclear envelope and because ANC-1 regulates ER positioning, we hypothesized that ANC-1 localizes to multiple membranes, including those away from the nuclear envelope. To study the localization of ANC-1, we tagged endogenous ANC-1 with GFP using CRISPR/Cas9 gene editing. The tag was placed either at the N-terminus of ANC-1b, or between the six tandem RPs and the F2 region to see if the opposite ends of ANC-1 localize to different structures (*Figure 5A*). Both strains were nearly wild type for hyp7 nuclear positioning (*Figure 5B*). Both GFP::ANC-1b and ANC-1::GFP::F2 localized in similar

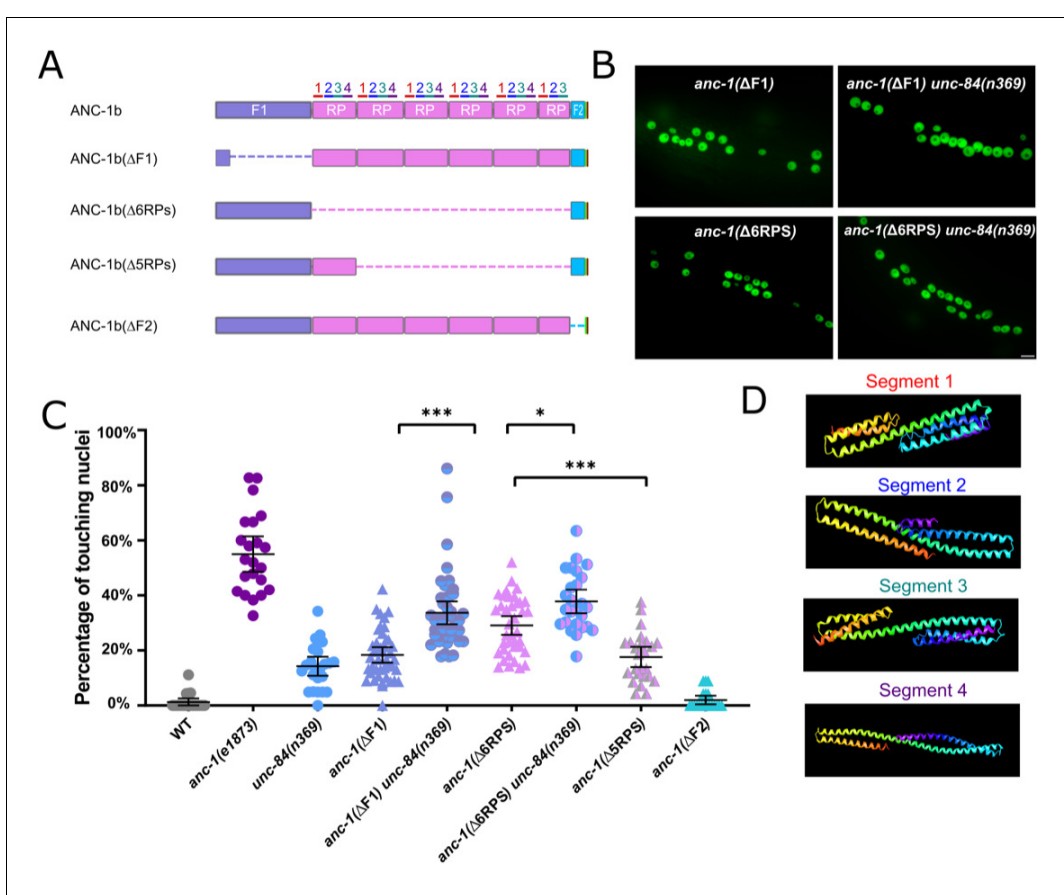

**Figure 3.** Cytoplasmic domain deletion analysis of ANC-1b. (A) Schematics of the ANC-1b cytoplasmic domain deletions. (B) Lateral views are shown of young adult *C. elegans* expressing hypodermal nuclear GFP in the indicated genotypes. Scale bar, 10 μm. (C) Quantification of nuclear anchorage in the wild type (WT) as well as *anc-1b* domain deletion mutants. Each point represents the percentage of touching nuclei on one side of a young adult animal. Means with 95% CI error bars are shown. ANOVA and Tukey's multiple comparisons tests were used for statistical analysis. *p≤0.05; **p≤0.01; ***p≤0.001. n ≥ 20 for each strain. The data in the first three columns, WT, *anc-1(e1873)*, and *unc-84(n369)* are duplicated from *Figure 1C* and copied here for easy reference. (D) QUARK result for three fragments in the tandem repeats (RPs). The positions of the fragments are indicated in 3A.

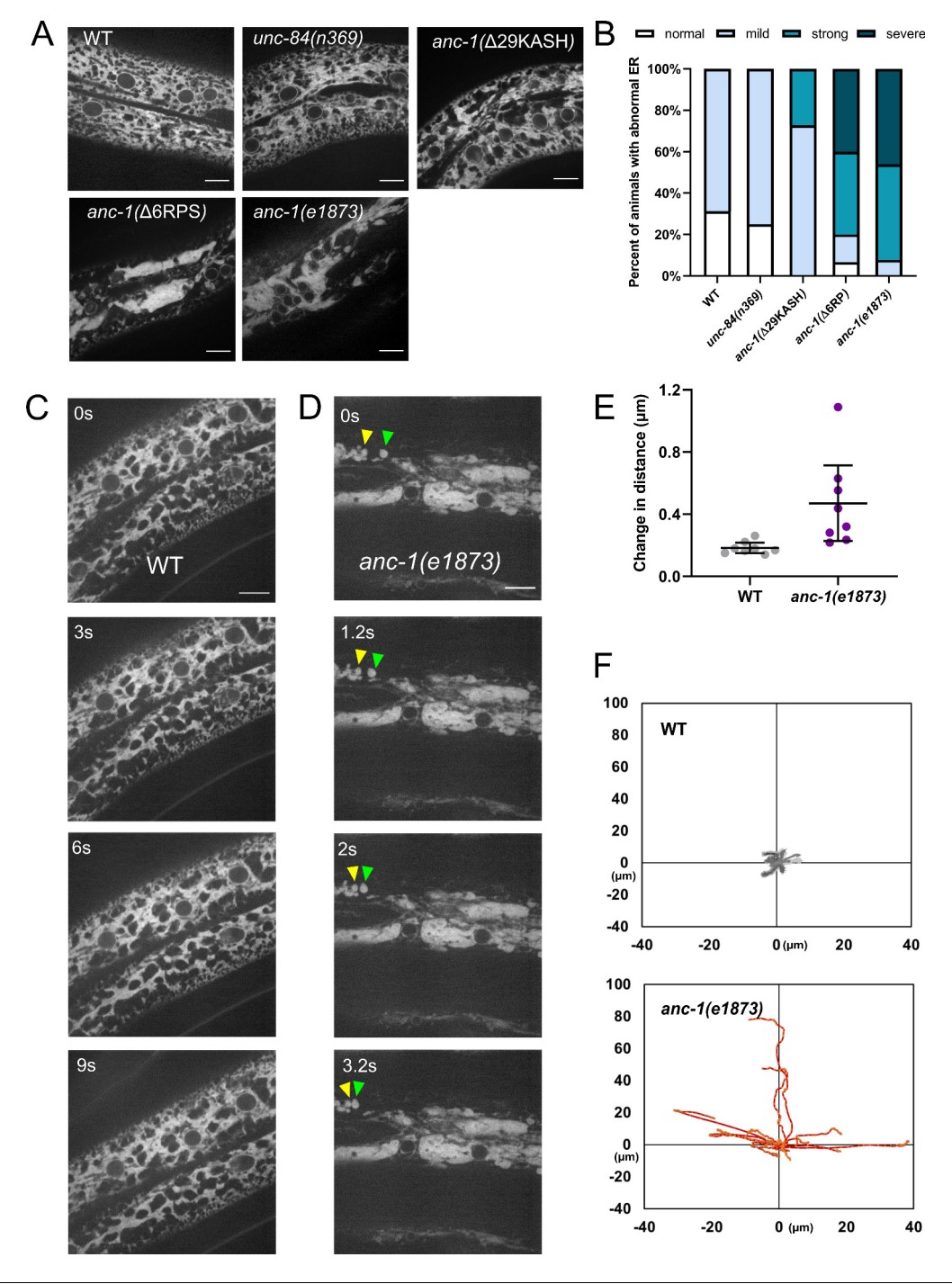

**Figure 4.** The ER is mispositioned in *anc-1* mutants. (A) Representative images of the hyp7 ER labeled with the GFP::KDEL marker in the young adult animals. (B) Scoring of the ER positioning defects. ER images of the listed strains were mixed and randomized for blind analysis by multiple researchers. n ≥ 11 for each strain. (C–D) Time-lapse images of hyp7 GFP::KDEL marker (C) over 9 s in wild type (WT) or (D) over 3.2 s in *anc-1* null. Arrowheads show two fragments of ER that changed their relative distance from one another over a short period of time. (E–F) To quantify ER displacement, three spots on each WT and *anc-1(e1873)* movie were tracked. The average change in distance between two points in 200 ms intervals is plotted in (E). The trajectories of the relative movements of each spot apart from the others are shown in (F). Eight movies of each strain were analyzed. Scale bar, 10 μm for all the images.

The online version of this article includes the following figure supplement(s) for figure 4:

*Figure 4 continued on next page*

*Figure 4 continued*

**Figure supplement 1.** Raw data for ER morphology assays.

patterns throughout the cytoplasm of adult hyp7 syncytia (*Figure 5C–D*). To examine the localization of the ANC-1b isoform alone, we introduced a premature stop codon mutation to disrupt the longer *anc-1a* and *c* isoforms in the GFP::ANC-1b strain, which did not significantly change GFP::ANC-1b localization (*Figure 5E*). These data are consistent with our model that *anc-1b* plays the major role in hyp7 nuclear positioning.

We next examined whether the cytoplasmic domains of ANC-1b are required for localization. Deletion of the F1 domain did not dramatically change localization of GFP::ANC-1b (*Figure 5F*). However, deletion of the six tandem repeats enriched GFP::ANC-1b around the nuclear envelope (*Figure 5G*), as did the deletion of five of the six repeats (*Figure 5H*). However, the intensity of the six repeat deletion mutant is significantly less than wild type GFP::ANC-1b (*Figure 5—figure supplement 1*), making it possible that the phenotype is due to less ANC-1 being present. Yet, the deletion mutant is significantly enriched at the nuclear envelope (*Figure 5—figure supplement 1B–C*), supporting the hypothesis that the defect is due to a loss of ANC-1 repeats at the general ER. The nuclear envelope enrichment of GFP::ANC-1b(Δ5RPs) was not observed when *unc-84* was mutated (*Figure 5I*), suggesting the nuclear envelope enrichment of GFP::ANC-1b is UNC-84-dependent.

The ANC-1 localization pattern in *Figure 6* is consistent with a model where ANC-1 localizes to ER membranes. In this model, a construct containing the transmembrane span near the C-terminus of ANC-1 but lacking the C-terminal, luminal part of the KASH domain would be sufficient for ER localization. Deletion of both the transmembrane span and the luminal parts of KASH with CRISPR/Cas9 resulted in a significantly worse nuclear anchorage defect than deleting only the luminal KASH domain (*Figure 6B*). Similarly, ER positioning defects were worse in *anc-1(ΔTK)* than in *anc-1 (Δ25KASH)* animals (*Figure 6C*). We next examined the role of the neck region, which is located immediately adjacent to the cytoplasmic side of the transmembrane span. When GFP was knocked-in between the neck region and the transmembrane span to make *anc-1(yc36[anc-1::gfp3Xflag::kash])*, it caused a significant nuclear anchorage defect (*Figure 6A–B*). Moreover, extending the deletion in *anc-1(ΔF2)*, which had no hyp7 nuclear anchorage defect (*Figure 3C*), an additional nine residues to remove the neck, caused a significant nuclear anchorage defect (*Figure 6A–B*). This suggests that the neck region next to the transmembrane span of ANC-1 plays a role in nuclear positioning, perhaps by targeting the C-terminus of ANC-1 to a membrane.

As shown above, a double mutant that lacks both the ANC-1 repeat region and LINC complex function in the form of an *unc-84(null)* had an intermediate defect (*Figure 3B*). This suggests that a third domain of ANC-1, in addition to the luminal portions of KASH and the RPs, is involved in positioning nuclei. To test this hypothesis, we made an *anc-1(gfp::anc-1b::Δ6rps::Δtk)* mutant line and found that it had a severe nuclear anchorage defect that was not significantly different from the *anc-1(e1873)* null mutant (*Figure 6B*). Together, these data support a model in which the transmembrane span is important for targeting ANC-1 to the ER/nuclear envelope membrane where ANC-1 then positions the ER and nuclei via LINC complex-independent mechanisms.

To further examine the role of the transmembrane span in ER and nuclear positioning, we

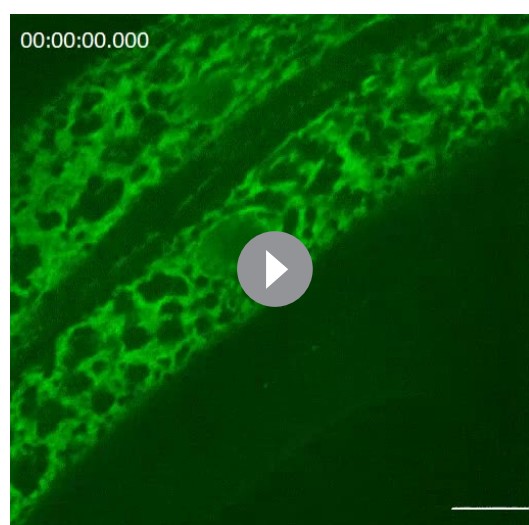

**Video 1.** The ER is anchored in wild type (WT) hyp7. An example video of the hyp7 ER in young adult wild type *C. elegans* expressing *pwSi83[p_hyp7_gfp::kdel]*. Images were captured at the interval of 0.2 s for 10 s. Scale bar, 10 μm.

https://elifesciences.org/articles/61069#video1

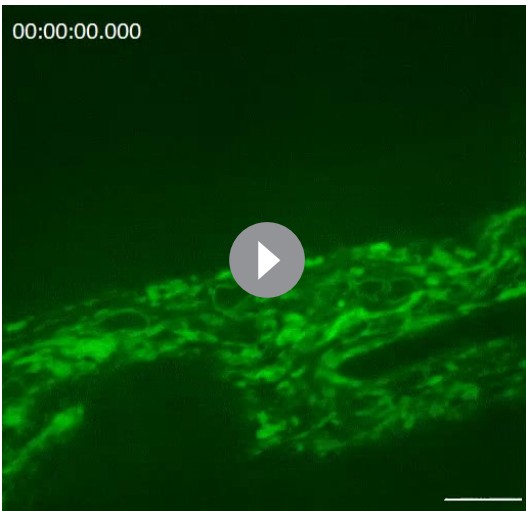

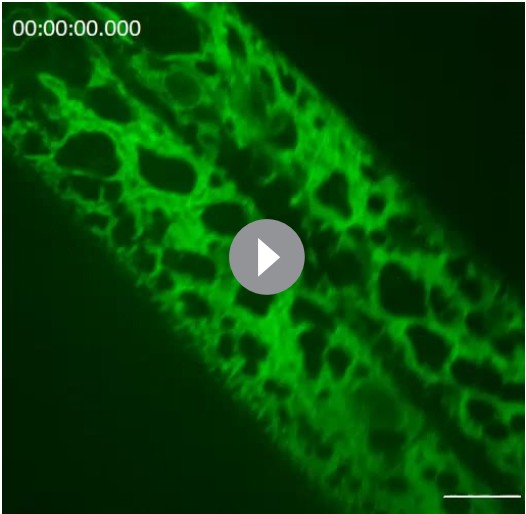

**Video 2.** The ER is unanchored in *anc-1(e1873)* mutant hyp7. An example video of the hyp7 ER in the young adult *anc-1(e1873)* mutant *C. elegans* expressing *pwSi83[p_{hyp7}gfp::kdel]*. Images were captured at the interval of 0.2 s for 10 s. Scale bar, 10 μm.
https://elifesciences.org/articles/61069#video2

**Video 3.** ER positioning in *unc-84(null)* mutant hyp7. An example of one of the most severe ER positioning defects observed is shown in a video of the hyp7 ER in the young adult *unc-84(n369)* mutant *C. elegans* expressing *pwSi83[p_{hyp7}gfp::kdel]*. Images were captured at the interval of 0.25 s for 4 s. Scale bar, 10 μm.
https://elifesciences.org/articles/61069#video3

observed the co-localization of ANC-1 with an ER membrane marker (*Figure 7*). A single-copy hypodermal-specific mKate2::TRAM-1 ER marker strain (*Rolls et al., 2002*) was generated and crossed with GFP::ANC-1b. The wild type GFP::ANC-1b fusion protein localized similarly to the ER marked by mKate2::TRAM-1 (*Figure 7A*). We used the ImageJ plug-in ScatterJ to quantify the co-localization and the Pearson's correlation coefficient averaged from 17 images (*Figure 7G*). Our second GFP construct, ANC-1::GFP::F2, also co-localized with the ER (*Figure 7B,G*). Similar results were obtained when co-localizing mKate2::ANC-1b and GFP::KDEL, but since the GFP::KDEL was overexpressed and significantly brighter than the mKate2::ANC-1b, we only present the analyses with GFP ANC-1 fusion proteins co-localizing with mKate2::TRAM-1, which is expressed at lower levels from a single-copy transgene. In contrast, GFP::ANC-1b did not co-localize with lipid droplets or mitochondria (*Figure 7—figure supplement 1*).

Deleting the luminal portion of the KASH domain from GFP::ANC-1b (GFP::ANC-1b::ΔKASH) did not significantly change its localization pattern relative to the wild type construct (*Figure 7C,G*). In contrast, deletion of both the luminal KASH peptide and the transmembrane span (GFP::ANC-1b::ΔTK) resulted in many cases where GFP::ANC-1b::ΔTK almost normally localized to the ER (*Figure 7D,G*) while in other animals it localized in parts of the cytoplasm that lacked ER (*Figure 7E, G*). Deletion of the repeat regions in GFP::ANC-1b::Δ6RPS led to a re-localization away from the general ER (*Figure 7F–G*). Together, these results suggest that (1) ANC-1b has a similar distribution pattern as the ER and (2) that the transmembrane span and repeat regions, but not the luminal KASH domain, plays roles ANC-1 ER localization. The ANC-1 localization pattern is consistent with the above findings that *anc-1(Δ6RPS) anc-1(ΔTK)* mutants had intermediate nuclear and ER positioning defects, worse than *anc-1(Δ25KASH)* mutants.

## Other organelles are mis-localized in *anc-1* mutants

Our imaging showed that both nuclei and the ER are both unanchored and move freely throughout the cytoplasm. Furthermore, our deletion analyses cast significant doubt on the old tethering model. In our new model, without ANC-1, the entire cytoplasm is disconnected and multiple organelles are likely flowing freely throughout the cytoplasm. We therefore used the MDT-28::mCherry (*Na et al., 2015*) marker to follow lipid droplets (*Figure 8*, *Videos 4–5*) and the mitoLS::GFP marker to follow mitochondria (*Figure 9*, *Videos 6–7*). In *anc-1* null mutant animals, both lipid droplets and

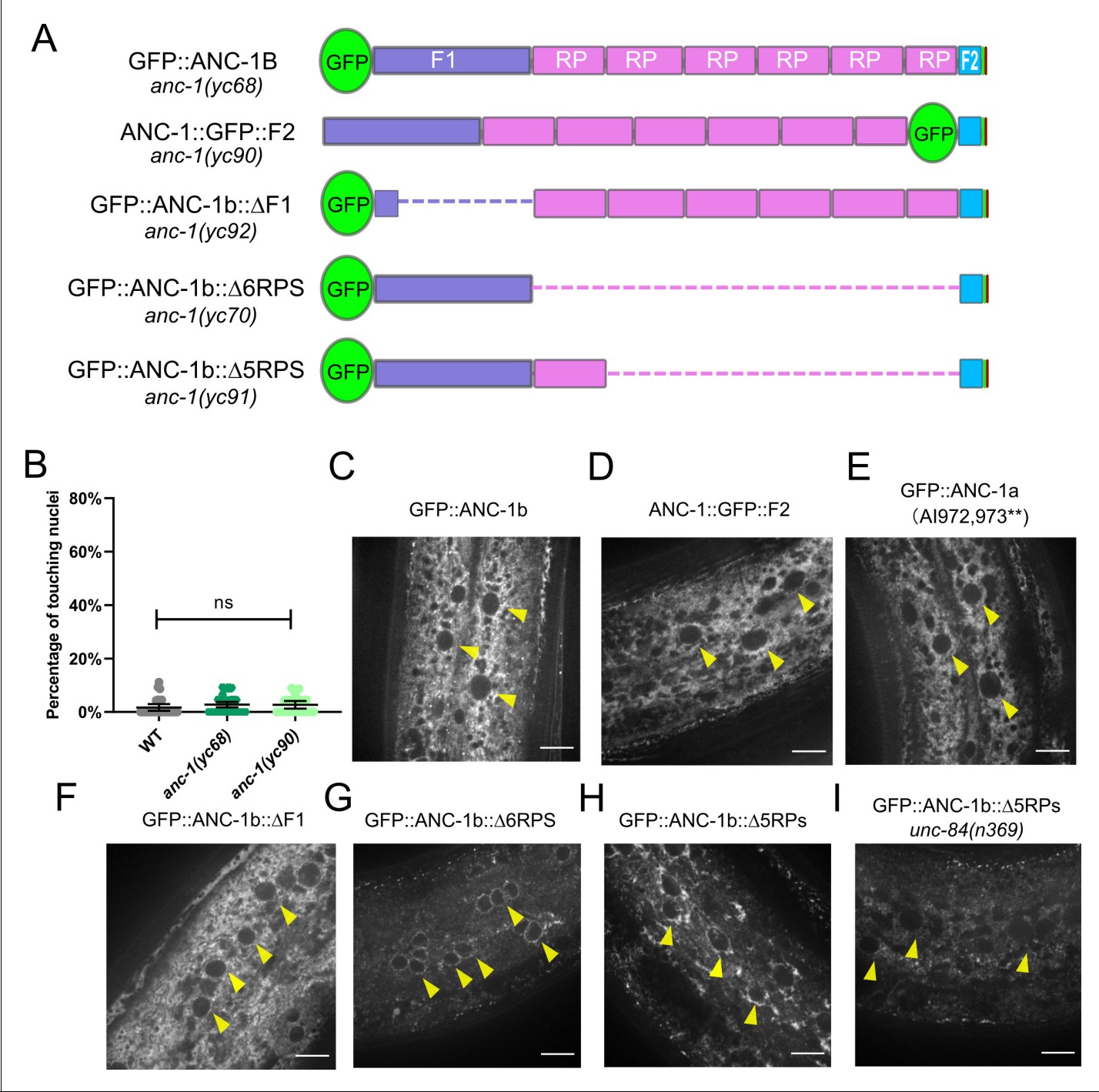

**Figure 5.** The subcellular localization of ANC-1. (A) Schematic depicting the ANC-1 GFP knock-in constructs with or without deletion of ANC-1 cytoplasmic domains. (B) Nuclear positioning in GFP::ANC-1b is wild type (WT). Each point represents the percentage of touching nuclei on one side of a young adult animal. Means with 95% CI error bars are shown. ns, not significant (p>0.05). n ≥ 20. (C–I) Confocal images of hyp7 subcellular localization in the indicated strains. Yellow arrowheads point to nuclei. Scale bar, 10 μm.

The online version of this article includes the following figure supplement(s) for figure 5:

**Figure supplement 1.** GFP::ANC-1(Δ6RPS) is expressed at a lower level and is enriched at the nuclear envelope.

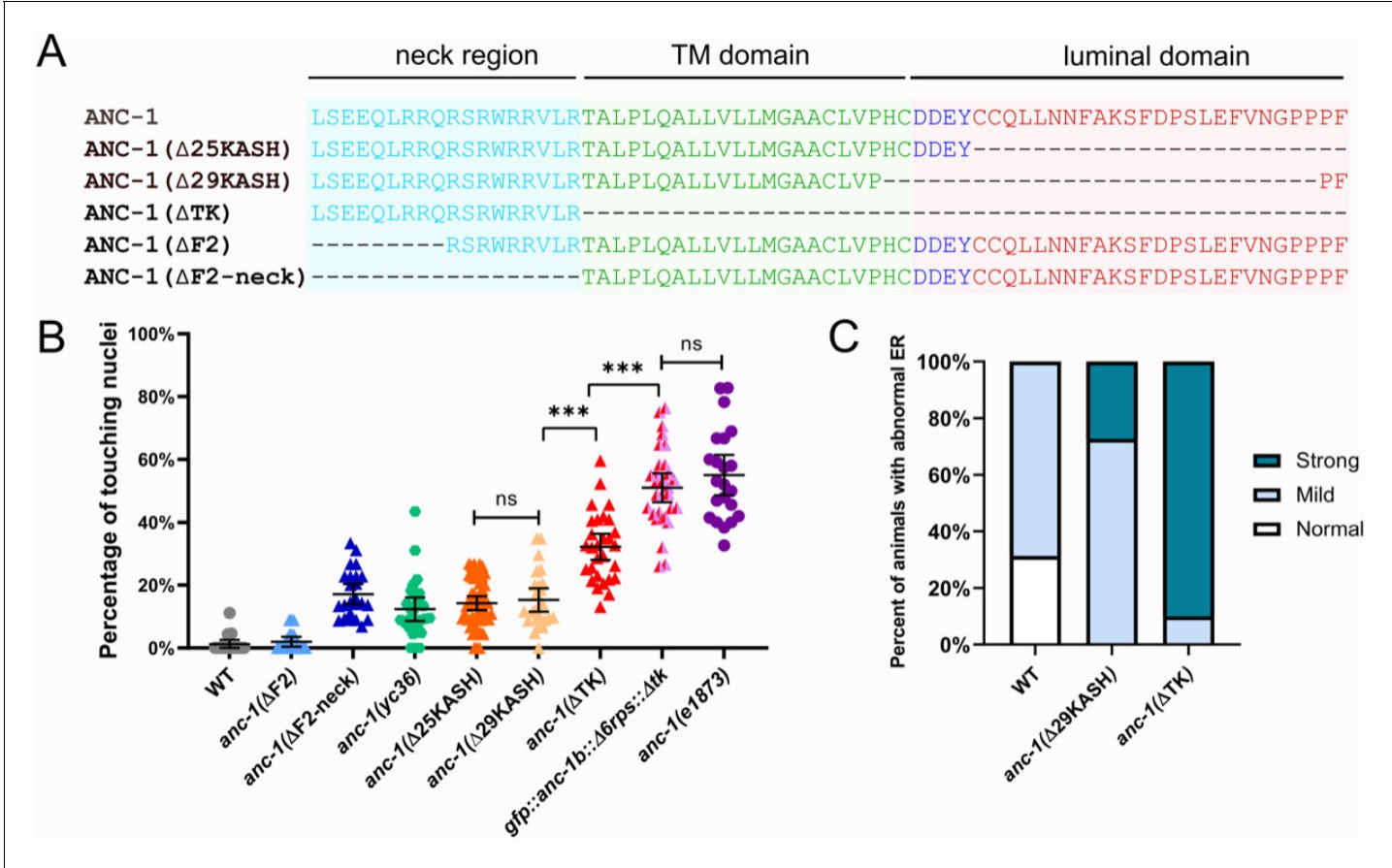

**Figure 6.** The transmembrane domain of ANC-1 is necessary for nuclear and ER positioning. (A) Sequence of deletion in the neck region (blue), transmembrane (TM) span (green), or the luminal domain (red). (B) Quantification of nuclear anchorage defects in *anc-1* mutants. Each point represents the percentage of touching nuclei on one side of a young adult animal. Means with 95% CI error bars are shown. n ≥ 20. ANOVA and Tukey's multiple comparisons tests were used in comparisons. ***p≤0.001; ns means p>0.05. The data of wild type (WT), *anc-1(ΔF2)*, *anc-1(Δ25KASH)*, *anc-1(Δ29KASH)*, and *anc-1(e1873)* are duplicated from *Figures 1C* and *3C* and copied here for easy reference. (C) Qualitative analysis of the ER anchorage defects as in *Figure 4*. Significantly more *anc-1(ΔTK)* animals show strong ER anchorage defects than *anc-1(Δ29KASH)* mutants (p≤0.01 by Fisher's exact test). Sample sizes were all ≥10.

mitochondria moved in a manner suggesting they were not connected to a network. Furthermore, lipid droplets were often seen in large clusters and mitochondria appeared slightly fragmented. We therefore conclude that ANC-1 is required to interconnect the entire cytoplasm to anchor nuclei, ER, lipid droplets, mitochondria, and likely other organelles to a single network that regulates the integrity of the cytoplasm.

## Microtubule networks appear normal in *anc-1* mutants

One hypothesis for ANC-1 function is that it regulates cytoskeletal networks. In this model, mutations in *anc-1* would disrupt the cytoskeleton, causing organelles to lose their attachment to the cytoskeleton and move around freely as shown in *Figures 4*, *8* and *9*. We therefore examined microtubule organization in the hyp7 syncytia of wild type and *anc-1* null mutant animals. We followed microtubules in live animals with an endogenously expressed microtubule binding protein, GFP::MAPH-1.1 (*Castiglioni et al., 2020*; *Waaijers et al., 2016*). The density and relative disorganization of the microtubule network in wild type made it difficult to quantify any growth, length, or number parameters of microtubules. Nonetheless, qualitatively, in comparison with the wild type (*Video 8*), microtubules were mostly normal in in *anc-1* null mutant animals (*Figure 10* and *Videos 9–10*). The exception was where nuclei had moved back and forth along an anterior-posterior axis, they appear to have cleared a microtubule free channel. These results are more in line with a model where in the

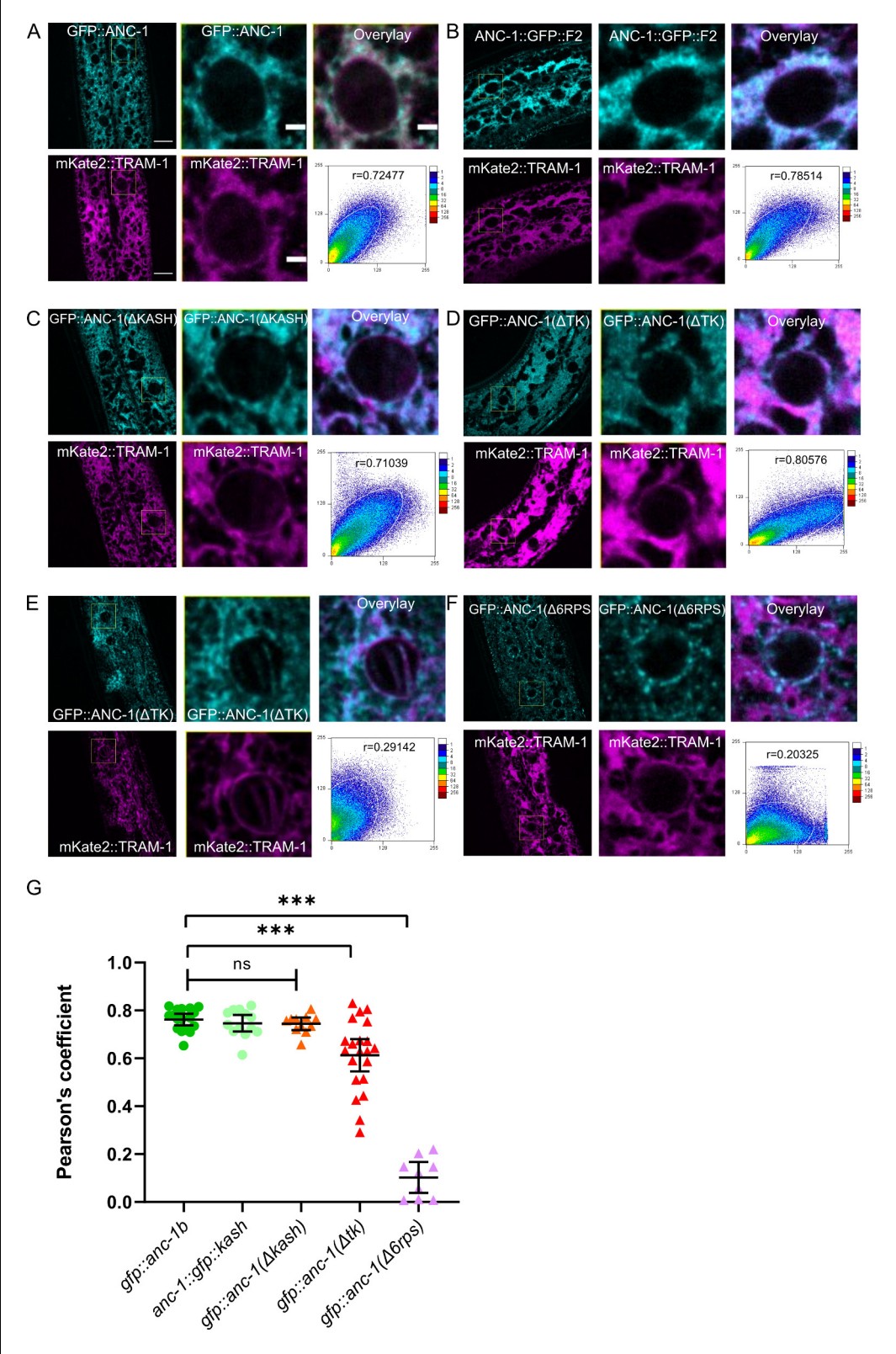

**Figure 7.** ANC-1 localizes to the ER. ANC-1-GFP fusion protein (cyan) localization with respect to the ER membrane as marked by a mKate::TRAM-1 (magenta) is shown. (**A**) Wild type (WT) GFP::ANC-1b. (**B**) Wild type ANC-1::GFP::F2. (**C**) GFP::ANC-1b::ΔKASH. (**D**) An example of GFP::ANC-1b::ΔTK with good overlap with the ER and (**E**) a GFP::ANC-1b::ΔTK with poor overlap. (**F**) GFP::ANC-1b::Δ6RPS. (**A–F**) For each section, the left two

*Figure 7 continued on next page*

*Figure 7 continued*

panels are low magnification of the young adult hypodermis (scale bar, 10 μm) and the middle two panels are a zoom in of the boxed part of the left panels (scale bar, 2 μm). The top right shows the merge of the two channels. The bottom right uses the used ImageJ plug-in ScatterJ to quantify the co-localization and the Pearson's correlation coefficient for overlap is shown as an r value. (G) A scatter plot of Pearson's coefficients showing overlap between the indicated GFP and the mKate::TRAM-1 ER membrane marker. Mean ±95% CI are shown. ANOVA and Tukey's multiple comparisons tests were used in comparisons. ***p≤0.001; ns means not significant. The online version of this article includes the following figure supplement(s) for figure 7:

**Figure supplement 1.** GFP::ANC-1b does not co-localize with lipid droplets or mitochondria.

absence of ANC-1, nuclei move around and disrupt or move microtubules in their way rather than a model where a lack of ANC-1 leads to massive microtubule disruption that then frees nuclei and other organelles to move throughout the syncytia.

## Depletion of ANC-1 disrupts nuclear morphology and causes developmental defects

Nuclear shape changes were observed during live imaging in *anc-1* mutants consistent with a model where *anc-1* mutant nuclei are susceptible to pressures from the cytoplasm, perhaps crashing into other organelles that corresponded with dents in nuclei. We therefore quantified nuclear size and

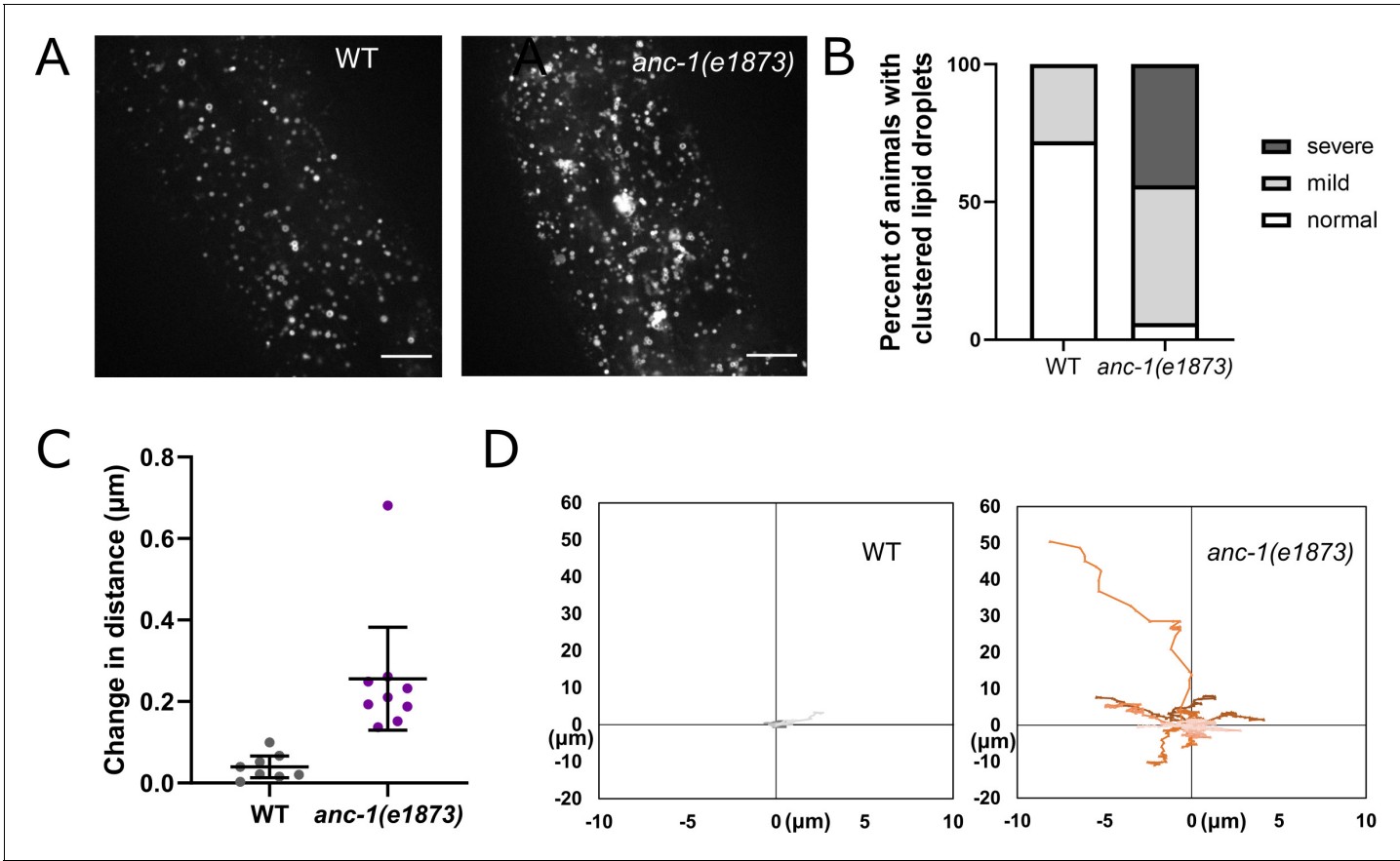

**Figure 8.** Lipid droplets are mispositioned in *anc-1* mutants. (A) Representative images of hyp7 lipid droplets labeled with the P*mdt-28*::*mdt-28*::*mCherry* marker in the young adult animals. (B) Scoring of the lipid droplet positioning defects. Images were randomized for blind analysis by multiple researchers. n ≥ 11. (C–D) Time-lapse images of hyp7 lipid droplets. (C–D) The lipid droplet displacement phenotype was quantified as described in *Figure 4E–F*. The average change in distance between two points in 242 ms intervals is plotted in (C). The trajectories of the relative movements of each spot apart from the others are shown in (D). Eight movies of each strain were analyzed. Scale bar, 10 μm for all the images. Also see *Videos 4–5*. WT = wild type.

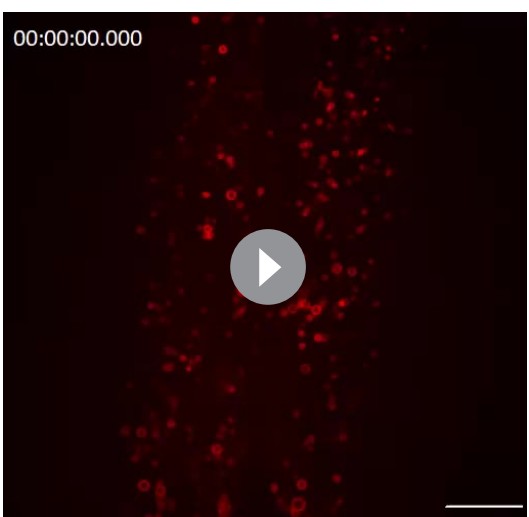

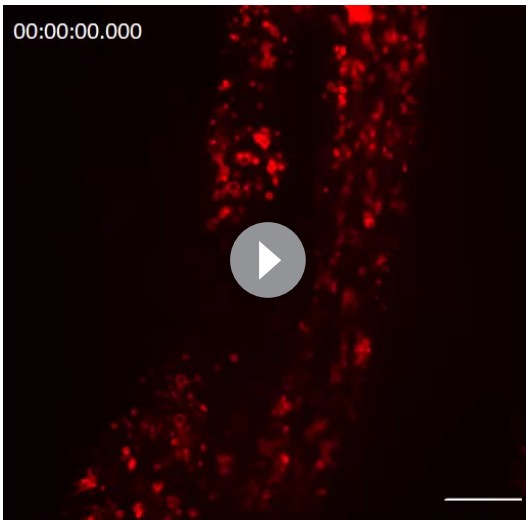

**Video 4.** Lipid droplets are anchored in wild type (WT) hyp7. An example video of the hyp7 lipid droplets in young adult wild type *C. elegans* expressing *ldrIs2 [mdt-28p::mdt-28::mCherry +unc-76(+)]*. Images were captured at the interval of 0.2 s for 10 s. Scale bar, 10 µm.
https://elifesciences.org/articles/61069#video4

**Video 5.** Lipid droplets are unanchored in *anc-1(e1873)* mutant hyp7. An example video of the hyp7 lipid droplets in the young adult *anc-1(e1873)* mutant *C. elegans* expressing *ldrIs2 [mdt-28p::mdt-28:: mCherry +unc-76(+)]*. Images were captured at the interval of 0.24 s for 10 s. Scale bar, 10 µm.
https://elifesciences.org/articles/61069#video5

shape in *anc-1* mutants to better characterize how nuclear and/or ER movements affect the nuclear structure. Adult syncytial hyp7 nuclei were significantly smaller in *anc-1(Δ6RPS)* and *anc-1(e1873)* mutants compared to wild type (*Figure 11A*). Furthermore, the shape of *anc-1(e1873)* hyp7 nuclei, as measured by circularity and solidity, was significantly less round than wild type (*Figure 11B*).

Given the nuclear anchorage, ER positioning, and nuclear shape defects observed in *anc-1* mutants, we examined whether these animals might have other developmental or growth defects. Despite producing fertile adults, *anc-1* null animals had severe developmental defects. The brood size of *anc-1(e1873)* mutants was less than 25% of wild type (*Figure 11C*) and *anc-1(e1873)* mutants had significantly smaller body sizes throughout larval and adult stages (*Figure 11D–F*). Together these results suggest that there are developmental consequences associated with organelle positioning defects in *anc-1* mutants.

## Discussion

LINC complexes consisting of giant KASH proteins (*C. elegans* ANC-1, *Drosophila* MSP300, and mammalian Nesprin-1 and -2) and canonical SUN proteins (*C. elegans* UNC-84, *Drosophila* Koi, and mammalian Sun1 and Sun2) are thought to anchor nuclei by tethering them to the cytoskeleton (*Starr and Fridolfsson, 2010*). ANC-1 was thought to connect actin filaments to the nuclear envelope using conserved CH and KASH domains at its N- and C-termini, respectively (*Starr and Han, 2002*). However, results discussed below indicate that this model is not sufficient to explain the role of KASH proteins in the cell. We propose a cytoplasmic integrity model, where giant KASH proteins function, largely independently of LINC complexes at the nuclear envelope and through mechanisms that do not require their CH domains. We find ANC-1 also localizes close to the ER throughout the cell. The large cytoplasmic domains of ANC-1 are required for positioning nuclei, ER, and likely other organelles. In the absence of ANC-1, the contents of the cytoplasm are disconnected from each other, unanchored in place, and sometimes fragmented. The cytoplasm and organelles freely flow throughout the hypodermal syncytia as worms crawl. It appears that ANC-1 is required to organize the entire cytoplasm.

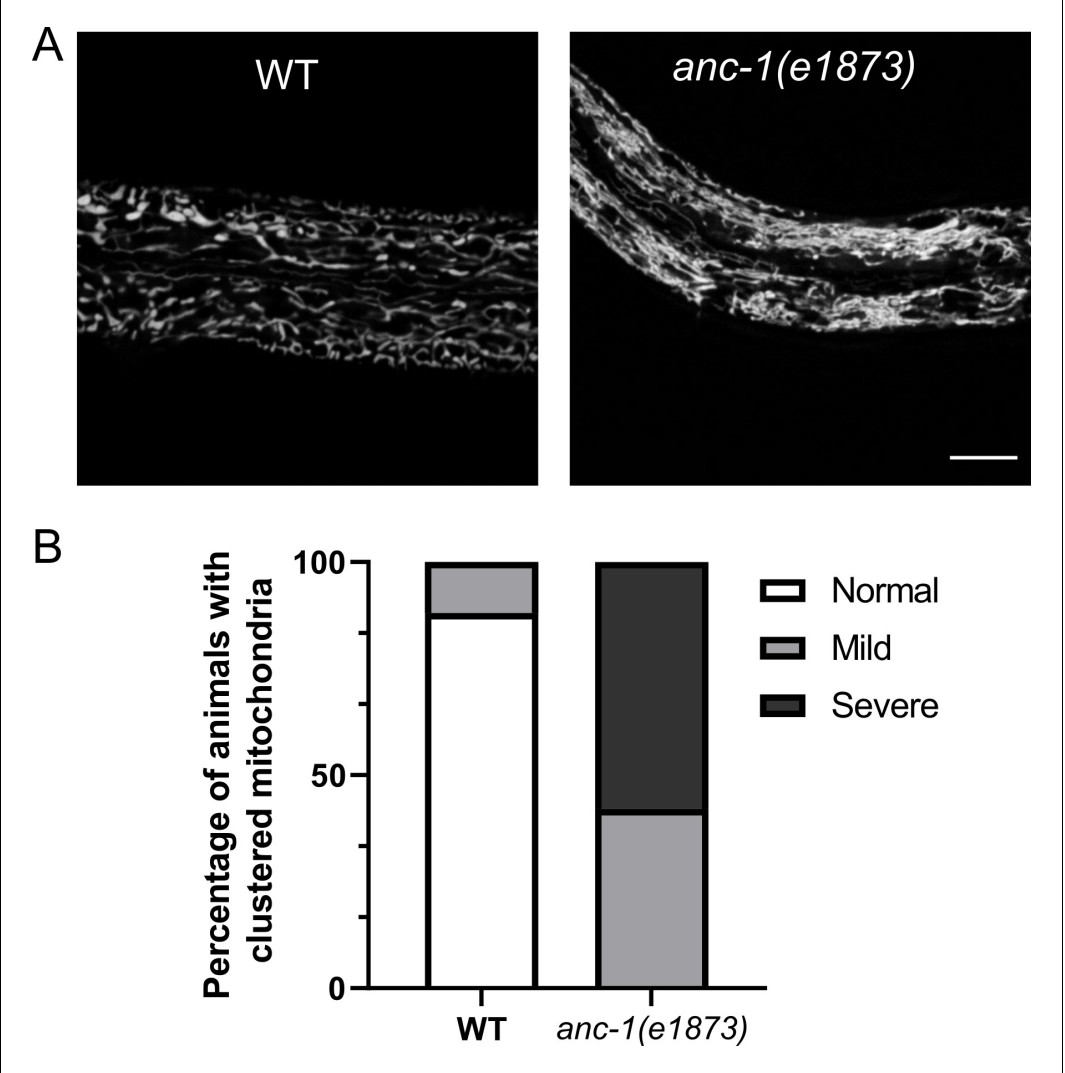

**Figure 9.** Mitochondria are mispositioned in *anc-1* mutants. (**A**) The hyp7 mitochondria were labeled with the $P_{col\text{-}10}$::*mito::GFP* marker in the L3 animals. (**B**) The mitochondria positioning defect was scored. Images were randomized for blind analysis by multiple researchers. n ≥ 18 for each strain. Scale bar, 10 μm for all the images. Also see *Videos 6–7*. WT = wild type.

Our finding that ANC-1 works independently of LINC complexes during hyp7 nuclear positioning has significant implications for how the field currently understands the role of LINC complexes in development and disease. A dominant negative approach relies on overexpression of KASH domains to displace endogenous KASH proteins from the nuclear envelope (*Grady et al., 2005*; *Starr and Han, 2002*; *Tsujikawa et al., 2007*). Alternatively, a mini-nesprin-2 construct, consisting of the calponin homology and KASH domains with a few spectrin repeats, is commonly used in rescue experiments (*Davidson et al., 2020*; *Luxton et al., 2010*; *Ostlund et al., 2009*). If ANC-1 and other giant KASH orthologs have major LINC complex-independent functions, these approaches might have more caveats than previously thought.

We found that the CH domain of ANC-1 is dispensable for ER and nuclear positioning. Similar findings suggest the Nesprin-1 or -2 CH domains are also dispensable for the development of mouse skeletal muscles and the epidermis (*Lüke et al., 2008*; *Stroud et al., 2017*). Moreover, the CH domains are not required for Nesprin-2 mediated neuronal migration in the developing rat brain (*Gonçalves et al., 2020*). Thus, the significance of the conserved CH domain is not clear. The CH domain of Nesprin-2 is required to form transmembrane actin-associated lines on the nuclear

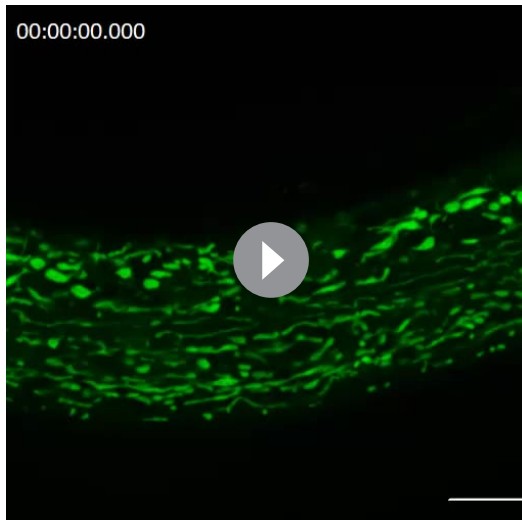

**Video 6.** Mitochondria are anchored in wild type (WT) hyp7. An example video of hyp7 mitochondria in young adult wild type *C. elegans* as followed by the $P_{col-10}$::*mitoLS::GFP* marker. Images were captured at the interval of 0.1 s for 10 s. Scale bar, 10 μm.
https://elifesciences.org/articles/61069#video6

envelope during rearward nuclear movement in mouse NIH3T3 fibroblasts polarizing for migration and to accumulate Nesprin-2 at the front of nuclei in cultured mouse embryonic fibroblasts migrating through constrictions (*Davidson et al., 2020*; *Luxton et al., 2010*).

Most of the other ANC-1 domains, including the C-terminal transmembrane span and the large cytoplasmic repeat regions, are required for nuclear anchorage. Portions of the tandem repeats may be arranged in bundles of three helices, reminiscent of spectrin repeats (*Liem, 2016*). Thus, we hypothesize that ANC-1, like the other giant KASH orthologs MSP300 and Nesprin-1 and -2, consists largely of spectrin repeats with a C-terminal transmembrane domain that attaches ANC-1 to the contiguous ER and outer nuclear membrane.

In general, giant KASH proteins localize strongly to the nuclear envelope in multiple systems (*Starr and Fridolfsson, 2010*). Evidence for giant KASH protein localization to other subcellular locations has been largely ignored or attributed to overexpression artifacts (*Zhang et al., 2001*) or the loss of KASH protein function (*Roux et al., 2009*). Antibodies against ANC-1 mostly localize away from nuclei and are only clearly enriched at the nuclear periphery in the somatic gonad (*Starr and Han, 2002*) and Nesprin-1 localizes to ciliary rootlets (*Potter et al., 2017*). Yet, most models, including the nuclear tethering model, focus primarily on nuclear envelope localization. We found that GFP::ANC-1b was not enriched at the nuclear envelope in most tissues (*Figure 5—*

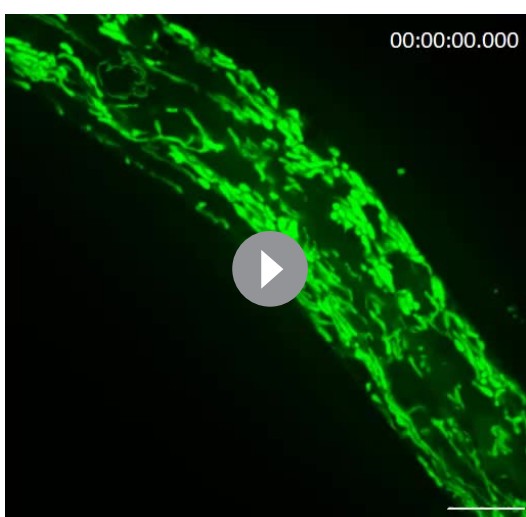

**Video 7.** Mitochondria are unanchored in *anc-1(e1873)* mutant hyp7. An example video of hyp7 mitochondria in the young adult *anc-1(e1873)* mutant *C. elegans* as followed by the $P_{col-10}$::*mitoLS::GFP* marker. Images were captured at the interval of 0.1 s for 10 s. Scale bar, 10 μm.
https://elifesciences.org/articles/61069#video7

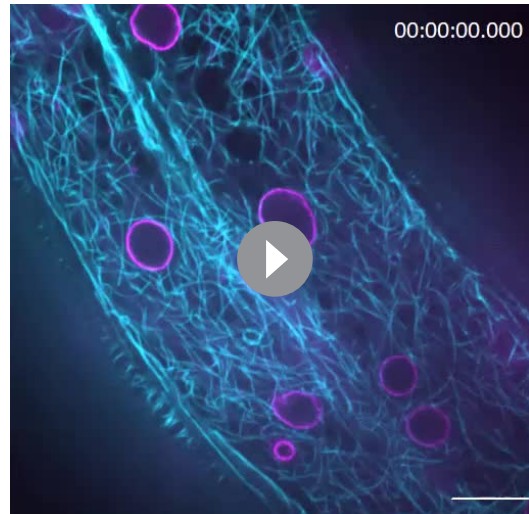

**Video 8.** Microtubules in wild type (WT) hypodermal syncytia. A representative clip of the hyp7 in a wild type young adult expressing GFP:: MAPH-1.1 to mark microtubules in cyan and EMR-1::mCherry to mark nuclear envelopes in magenta. Images were at the interval of 1 s for 20 s. Scale bar, 10 μm.
https://elifesciences.org/articles/61069#video8

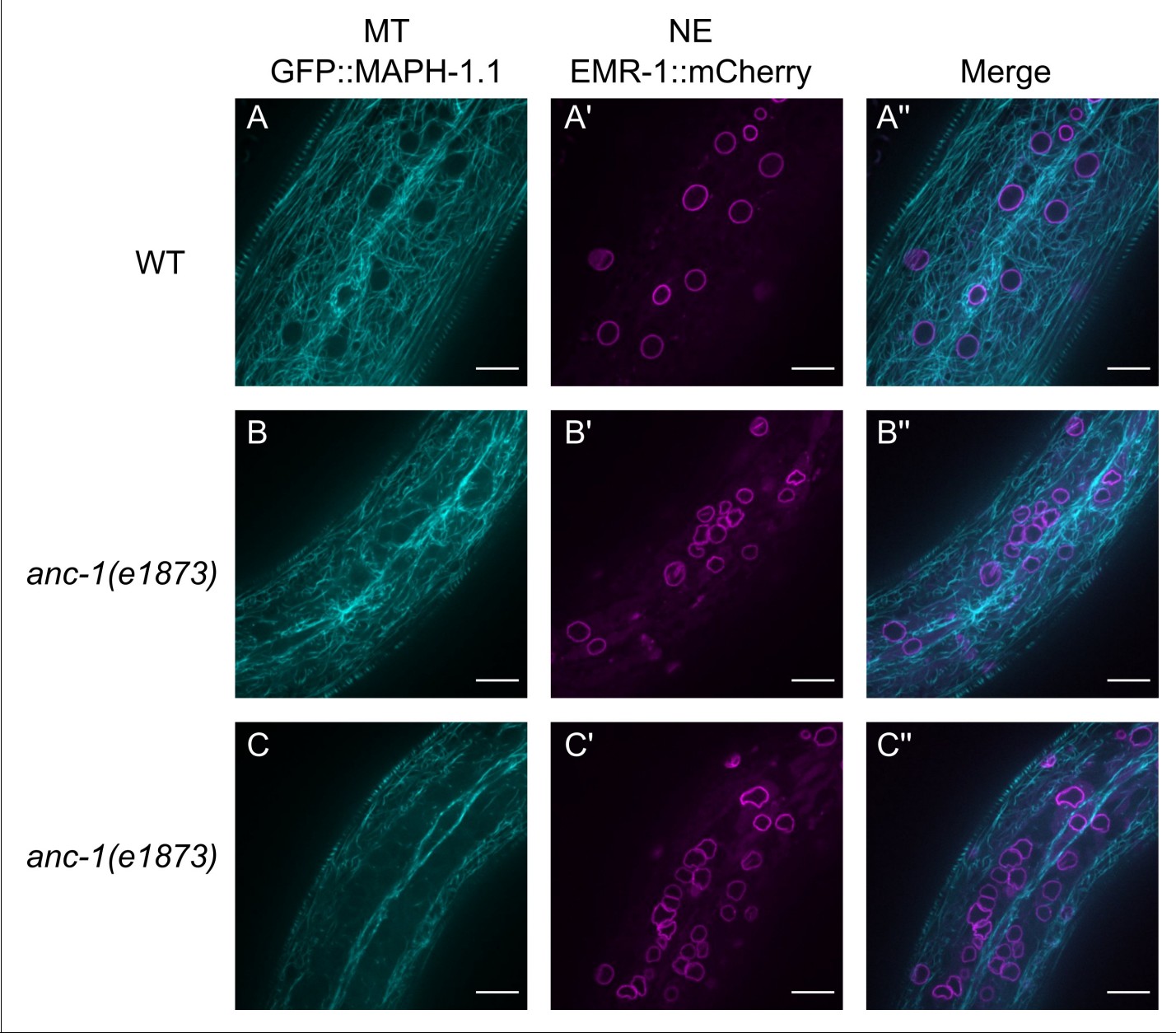

**Figure 10.** Microtubule organization in *anc-1* mutants. Representative images of microtubules (cyan) and nuclear envelope (NE) (magenta) from the same young adult animals are shown. (A–A'') wild type (WT), (B–B'') *anc-1(e1873)* mutant animal where the microtubules are mostly normal. (C–C'') In other *anc-1(e1873)* animals, microtubule organization is disrupted in the channels where nuclei move. Microtubules are labeled with GFP::MAPH-1.1 and the nuclear envelope is labeled with EMR-1::mCherry. Scale bar, 10 μm. Also see *Videos 8–10*.

*figure supplement 1*). Instead, our findings suggest that ANC-1 functions throughout the cytoplasm with its C-terminus in the ER membrane.

We observed that the ER is severely mispositioned in *anc-1* null mutant hypodermal cells. Also, mitochondria are misshaped and mispositioned in the *anc-1* null mutants, but not in *unc-84* nulls (*Hedgecock and Thomson, 1982*; *Starr and Han, 2002*). Two different mechanisms could explain how nuclei, mitochondria, and the ER are all mispositioned in *anc-1* mutants. First, abnormal organelle clusters could result from a failure in the active positioning process (*Folker and Baylies, 2013*; *Roman and Gomes, 2018*). Alternatively, organelles could lose physical connections to the rest of the cytoplasm, resulting in organelles being passively pushed around in response to external mechanical forces, such as those generated when the worm crawls. Live imaging favors the second

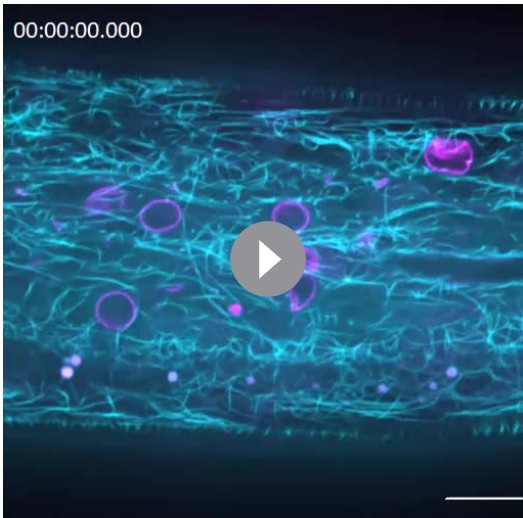

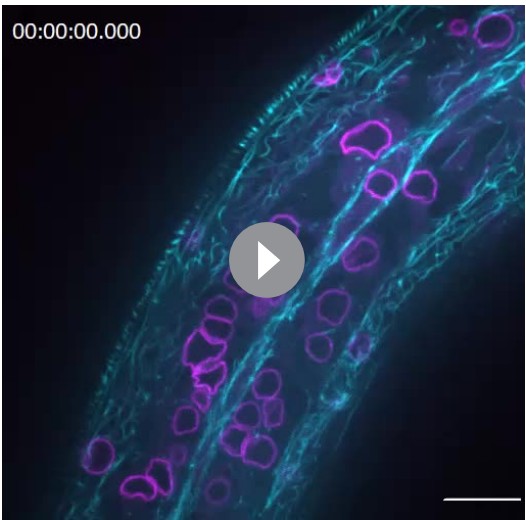

**Video 9.** Microtubules in *anc-1(null)* hypodermal syncytia. A representative clip of the hyp7 in a *anc-1 (e1783)* mutant young adult expressing GFP:: MAPH-1.1 to mark microtubules in cyan and EMR-1::mCherry to mark nuclear envelopes in magenta. Images were at the interval of 0.845 s for 40 s. Scale bar, 10 μm.
https://elifesciences.org/articles/61069#video9

**Video 10.** Microtubules in *anc-1(null)* hypodermal syncytia. A representative clip of the hyp7 in a *anc-1 (e1783)* mutant young adult expressing GFP::MAPH-1.1 to mark microtubules in cyan and EMR-1::mCherry to mark nuclear envelopes in magenta. Images were at the interval of 0.638 s for 32 s. Scale bar, 10 μm.
https://elifesciences.org/articles/61069#video10

model. As *anc-1* mutant worms bent their bodies, ER fragments rapidly moved in a bulk flow of cytoplasm along with nuclei. The nuclei became misshapen and the ER often fragmented. Nuclei and the ER appeared to be completely detached from any sort of cytoplasmic structural network in *anc-1* mutants. There is precedent that giant KASH proteins could mechanically stabilize the cytoplasm. In cultured mouse NIH3T3 fibroblasts, disrupting KASH proteins leads to the decreased stiffness across the entire cell, as determined by single-particle microrheology experiments (*Stewart-Hutchinson et al., 2008*). Therefore, we propose that ANC-1 and other giant KASH proteins normally function at the ER periphery as important crosslinkers to maintain the mechanical integrity of the cell.

ANC-1 likely maintains the mechanical integrity of the cytoplasm via cytoskeletal networks. Disruption of MSP300 leads to disruption of both actin and microtubule networks in *Drosophila* muscles (*Wang et al., 2015*). Giant KASH proteins can interact with actin independently of their CH domains. ANC-1 binds to AMPH-1/BIN1, which is involved in actin nucleation (*D'Alessandro et al., 2015*), and Nesprin-2 binds to the actin regulators Fascin and FHOD1 (*Jayo et al., 2016*; *Kutscheidt et al., 2014*). Nesprin-1 interacts with Akap450 to form microtubule organizing centers at the nuclear envelope of mouse muscles (*Elhanany-Tamir et al., 2012*; *Gimpel et al., 2017*; *Zheng et al., 2020*). Intermediate filaments and spectraplakins have also been implicated in nuclear positioning (*Ralston et al., 2006*; *Wang et al., 2015*; *Zheng et al., 2020*). However, *anc-1* null mutant animals had a reasonably normal microtubule network. Thus, how ANC-1b might directly interact with cytoskeletal components requires further investigations.

In summary, we propose a cytoplasmic integrity model, for how ANC-1 and giant KASH orthologs position nuclei and other organelles. First, ANC-1 is targeted to the ER through its C-terminal transmembrane span. The large cytoplasmic domain of ANC-1 likely consists of divergent spectrin-like repeats that could serve as elastic filaments. ANC-1 filaments could interact with various components of the cytoskeleton, allowing ANC-1 to serve as a cytoskeletal crosslinker to maintain the mechanical integrity of the cytoplasm. However, since at least microtubules are not completely disrupted, ANC-1 likely functions to maintain the mechanical properties of the cytoplasm through other mechanisms. Extracellular mechanical pressure that is generated as the worm bends leads to rapid cytoplasmic fluid flows and positioning defects in nuclei, ER, mitochondria, and likely other

organelles. In this model, ANC-1 maintains cellular connectivity across the cytoplasm by anchoring organelles in place.

# Materials and methods

## Key resources table

| Reagent type (species) or resource | Designation | Source or reference | Identifiers | Additional information |
|---|---|---|---|---|
| Strain, strain background (*Escherichia. coli*) | OP50 | Caenorhabditis Genetics Center (CGC) | OP50 | https://cgc.umn.edu/strain/OP50 |
| Strain, strain background (*E. coli*) | DH10B | New England Biolabs (NEB) | C3019H | |
| Recombinant DNA reagent | pLF3FShC | *Nonet, 2020* | addgene: #153083 | https://www.addgene.org/153083/ |
| Recombinant DNA reagent | pSL845 | This paper | | $P_{y37a1b.5}$::mKate2::tram-1::tram-1 3'UTR plasmid to generate the *ycSi2* transgenic *C. elegans* strain |
| Recombinant DNA reagent | pSL835 | This paper | | $P_{anc-1b}$::nls::gfp::lacZ Plasmid to generate *ycEx260* |
| Recombinant DNA reagent | pSL289 | This paper | | $P_{col-10}$::mitoLS::gfp Plasmid to generate *ycEx217* |

## *C. elegans* genetics

*C. elegans* strains were maintained on nematode growth medium plates seeded with OP50 *E. coli* at the room temperature (approximately 22℃) (*Brenner, 1974*). All the strains used in this study are listed in *Table 1*. Some strains, including N2 (WB Cat# WBStrain00000001, RRID:WB-STRAIN: WBStrain00000001), which was used as wild type, were obtained from the Caenorhabditis Genetics Center, funded by the National Institutes of Health Office of Research Infrastructure Programs (P40 OD010440). Strains VC40007, VC20178, and VC40614 were provided by the *C. elegans* Reverse Genetics Core Facility at the University of British Columbia (*Thompson et al., 2013*). Strain RT3739 (*pwSi83*) was generously provided by Barth Grant (Rutgers University, NJ, USA). UD522 (*ycEx249* [*pcol-19::gfp::lacZ, pmyo-2::mCherry*]) was previously described (*Cain et al., 2018*). Male strains of RT3739, UD522, BOX188, LIU2, UD756, UD3, and BN147 were made to cross into *anc-1* or *unc-84* mutants. Alternatively, *pcol-19::gfp::lacZ* was introduced into some mutants by standard germline transformation to make UD736 and UD737 (*Cain et al., 2018*; *Mello et al., 1991*).

For *anc-1* RNAi feeding experiments, L4 stage animals were transferred onto NGM plates seeded with bacteria expressing dsRNA (*Timmons and Fire, 1998*). The clones from the Ahringer RNAi library (Source Bioscience) (*Kamath and Ahringer, 2003*) were confirmed by Sanger sequencing.

Nuclear anchorage assays were performed as described (*Cain et al., 2018*; *Fridolfsson et al., 2018*). Briefly, L4 worms with GFP-marked hypodermal nuclei were picked onto fresh plates 20 hr before scoring. Young adults were mounted on 2% agarose pads in ~5 µl of 1 mM tetramisole in M9 buffer (*Fridolfsson et al., 2018*). Syncytial hyp7 nuclei were scored as touching if a nucleus was in contact with one or more neighboring nuclei. Only one lateral side of each animal was scored.

For the brood size assay, starting at the L4 stage, single animals were transferred onto fresh OP50 *E. coli* plates every 24 hr for seven days. The number of embryos laid were counted immediately after the removal of the animal each day.

To measure body size, embryos laid within an hour were collected and cultured for 24 hr and 69 hr to reach the L2 stage and the adult stage, respectively. Animals were mounted on 2% agarose pads in ~5 µl of 1 mM tetramisole in M9 buffer for imaging.

## 5'-Rapid amplification of cDNA ends (5'-RACE)

Total RNA was extracted from mixed stages of *C. elegans* using the RNeasy kit (QIAGEN). First-strand cDNAs were generated with the ThermoScript RT-PCR system using an *anc-1* antisense

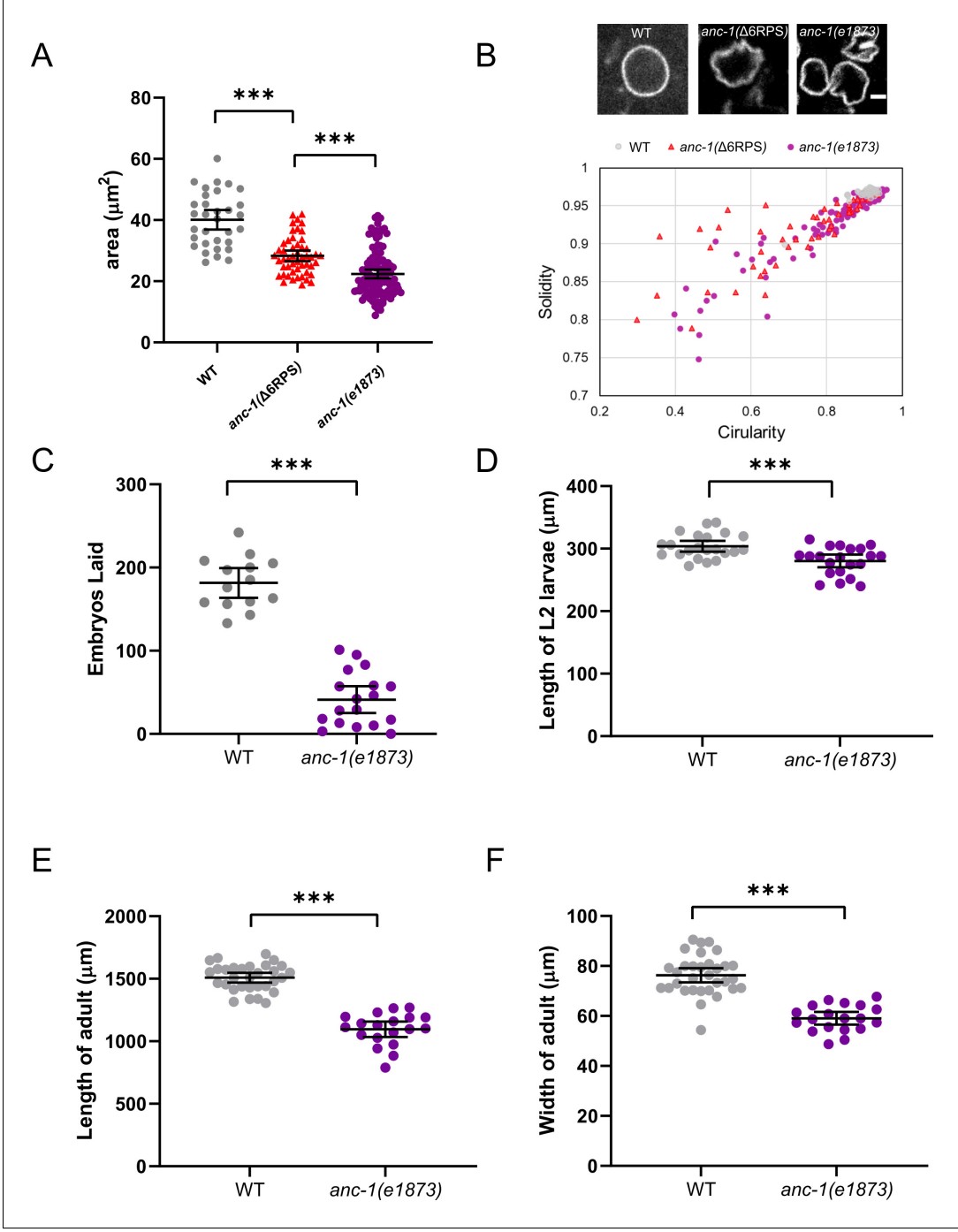

**Figure 11.** *anc-1* mutants have developmental defects. (**A**) The area of cross-sections of hyp7 nuclei is shown. Each dot represents the area of a single nucleus. n = 32 for wild type (WT), n = 50 for *anc-1(Δ6RPS)*, n = 111 for *anc-1(e1873)*. (**B**) Top panel: Representative images of hyp7 nuclei marked by EMR-1::mCherry of WT, *anc-1* (*Δ6RPS*), and *anc-1(e1873)* mutants. Scale bar, 2 μm. Bottom panel: Plot of the solidity and the circularity. (**C–F**) The brood size (**C**), length of the L2 larvae (**D**), adult length (**E**) and width (**F**) are significantly reduced in *anc-1(e1873)* mutants. Each dot represents a single animal. n ≥ 14 for (**C**) and n ≥ 19 for (**D–F**). Means with 95% CI error bars are shown. Unpaired student two-tail t-test was used for statistical analysis. \*\*p≤0.01; \*\*\*p≤0.001.

oligonucleotide (ods2572: 5'-ATAGATCATTACAAGATG-3'). Purification and TdT tailing of the first-strand cDNA were performed by the 5' RACE System for Rapid Amplification of cDNA Ends, version 2.0 (Invitrogen, Cat. No. 18374058). The target cDNA was amplified PCR using the provided 5'

**Table 1.** *C. elegans* strains in this study.

| Strain | Genotype | Reference |
| --- | --- | --- |
| N2 | Wild type | |
| UD522 | ycEx249[p_{col-19}::gfp::lacZ, p_{myo-2}::mCherry] | *Cain et al., 2018* |
| UD532 | unc-84(n369) X; ycEx249 | *Cain et al., 2018* |
| UD538 | anc-1(e1873) I; ycEx249 | *Cain et al., 2018* |
| UD578 | anc-1(yc52[Δ25KASH]) I; ycEx249 | This study |
| UD615 | anc-1(yc69[Δ29KASH]) I; ycEx249 | This study |
| JR672 | wIs54[scm::gfp] V | *Terns et al., 1997* |
| UD457 | unc-84(n369) X; wIs54 V | This study |
| UD451 | anc-1(e1873) I; wIs54 V | This study |
| VC20178 | anc-1(gk109010[W427*]) I | *Thompson et al., 2013* |
| UD737 | anc-1(gk109010[W427*]) I; ycEx 265[p_{col-19}::gfp::lacZ, p_{myo-2}::mCherry] | This study |
| VC40007 | anc-1(gk109018[W621*]) I | *Thompson et al., 2013* |
| UD736 | anc-1(gk109018[W621*]) I; yc Ex266[p_{col-19}::gfp::lacZ, p_{myo-2}::mCherry] | This study |
| VC40614 | anc-1(gk722608[Q2878*]) I | *Thompson et al., 2013* |
| UD565 | anc-1(gk722608[Q2878*]) I; ycEx249 | This study |
| UD535 | anc-1(yc41[anc-1::gfp3Xflag::kash,Δch]) I; ycEx249 | This study |
| UD608 | ycEx260[p_{anc-1b}::nls::gfp::lacZ] | This study |
| UD599 | anc-1(yc62[ΔF1]) I; ycEx249 | This study |
| UD591 | anc-1(yc61[Δ6RPS]) I; ycEx249 | This study |
| UD668 | anc-1(yc80[ΔF2]) I; ycEx249 | This study |
| UD669 | anc-1(yc81[ΔF2-neck]) I; ycEx249 | This study |
| UD695 | anc-1(yc61[Δ6RPS]) I; unc-84(n369) X; ycEx249 | This study |
| UD696 | anc-1(yc62[ΔF1]) I; unc-84(n369) X; ycEx249 | This study |
| UD612 | anc-1(yc68[gfp::anc-1b]) I | This study |
| UD655 | anc-1(yc78[Δ5RPS]) I; ycEx249 | This study |
| UD619 | anc-1(yc68[gfp::anc-1b]) I; ycEx249 | This study |
| UD694 | anc-1(yc90[anc-1::gfp::F2]) I | This study |
| UD697 | anc-1(yc90[anc-1::gfp::F2]) I; ycEx249 | This study |
| UD618 | anc-1(yc70[gfp::anc-1b::Δ6rps]) I | This study |
| UD698 | anc-1(yc91[gfp::anc-1b::Δ5rps]) I | This study |
| UD701 | anc-1(yc92[gfp::anc-1b::Δf1]) I | This study |
| UD702 | anc-1(yc93[anc-1AI972,973**::gfp]) I | This study |
| UD625 | anc-1(yc71[ΔTK]) I; ycEx249 | This study |
| UD645 | anc-1(yc73[gfp::anc-1b::Δ6rps::Δtk]) I; ycEx249 | This study |
| RT3739 | pwSi83[p_{hyp7}gfp::kdel] | A gift from Barth Grant |
| UD652 | anc-1(e1873) I; pwSi83 | This study |
| UD679 | unc-84(n369) X; pwSi83 | This study |
| UD707 | anc-1(yc71[ΔTK]) I; pwSi83 | This study |
| UD651 | anc-1(yc61[Δ6RPS]) I; pwSi83 | This study |
| UD672 | anc-1(yc69[Δ29KASH]) I; pwSi83 | This study |
| UD520 | anc-1(yc36[anc-1::gfp3Xflag::kash]) I | This study |

*Table 1 continued on next page*

*Table 1 continued*

| Strain | Genotype | Reference |
|---|---|---|
| BN147 | bqSi142 [p_emr-1_::emr-1::mCherry + unc-119(+)] II | *Morales-Martínez et al., 2015* |
| BOX188 | maph-1.1(mib12[gfp::maph-1.1]) I | *Waaijers et al., 2016* |
| UD649 | maph-1.1(mib12[gfp::maph-1.1]) I; bqSi142 II | This study |
| UD650 | anc-1(e1873) I; maph-1.1(mib12[gfp::maph-1.1]) I; bqSi142 II | This study |
| UD677 | anc-1(yc61[Δ6RPS]) I; maph-1.1(mib12[gfp::maph-1.1]) I; bqSi142 II | This study |
| UD3 | ycEx217[P_col-10_::mitoLS::GFP, P_odr-1_::RFP] | This study |
| UD676 | anc-1(e1873) I; [P_col-10_::mitoLS::GFP, P_odr-1_::RFP] | This study |
| LIU2 | ldrIs2 [mdt-28p::mdt-28::mCherry + unc-76(+)]. | *Na et al., 2015* |
| UD681 | anc-1(e1873) I; ldrIs2 | This study |
| UD728 | anc-1(yc94[gfp::anc-1b::Δtk]) I | This study |
| UD789 | anc-1(yc106[gfp::anc-1b::Δkash]) I | This study |
| UD756 | ycSi2[pSL845 P_y37a1b.5_::mKate2::tram-1::tram-1 3'UTR] IV | This study |
| NM5179 | jsTi1493[LoxP::mex-5p::FLP:SL2::mNeonGreen::rpl-28p::FRT::GFP::his-58::FRT3] IV | *Nonet, 2020* |
| UD778 | anc-1(yc68[gfp::anc-1b]) I; ycSi2 | This study |
| UD779 | anc-1(yc90[anc-1::gfp::F2]) I; ycSi2 | This study |
| UD780 | anc-1(yc70[gfp::anc-1b::Δ6rps]) I; ycSi2 | This study |
| UD781 | anc-1(yc106[gfp::anc-1b::Δkash]) I; ycSi2 | This study |
| UD782 | anc-1(yc94[gfp::anc-1b::Δtk]) I; ycSi2 | This study |
| UD783 | anc-1(yc68[gfp::anc-1b]) I; bqSi142 | This study |
| UD785 | anc-1(yc70[gfp::anc-1b::Δ6rps]) I; bqSi142 | This study |
| UD790 | anc-1(yc68[gfp::anc-1b]) I; ldrIs2 | This study |

RACE Abridged Anchor Primer and an *anc-1* specific primer: ods2574 (5'-GTCGGCGTCTGAAG-GAAAGA-3'). The PCR product was purified using the QIAquick PCR purification kit (QIAGEN) and Sanger sequencing was performed by Genewiz.

## Plasmid construction and transformation

To generate plasmid $p_{anc-1b}$::nls::gfp::lacZ (pSL835), a 2.56 kb fragment of genomic DNA upstream of the start codon of *anc-1b* was amplified with primers ods2491 (5'- TACCGAGCTCAGAAAAAA TGACTGTGAGTATAGTCATTTTCCGCT-3') and ods2492 (5'-GTACCTTACGCTTCTTC TTTGGAGCCATTTTGGTTCGGAGCAC-3') to replace the *col-19* promoter of $p_{col-19}$::gfp::lacZ (pSL779) (*Cain et al., 2018*). N2 animals were injected with 45 ng/µl of pSL835, 50 ng/µl of pBlue-script SK, and 2.5 ng/µl of pCFJ90 ($p_{myo-2}$::mCherry) (*Frøkjaer-Jensen et al., 2008*) by standard *C. elegans* germline transformation (*Evans, 2006*) to make strain UD608 (*ycEx260[Panc-1b::nls::gfp:: lacZ-2]*). Plasmid $P_{col-10}$::mitoLS::gfp (pSL289) and the odr-1::rfp co-injection marker were injected into N2 young adults to generate UD3 *ycEx217[P_col-10_::mitoLS::GFP, Podr-1::RFP]* transgenic strain. Flp Recombinase-Mediated Cassette Exchange (RMCE) method was used to generate strain UD756 *ycSi2[pSL845 P_y37a1b.5_::mKate2::tram-1::tram-13'UTR]* (*Nonet, 2020*). pLF3FShC was a gift from Michael Nonet (Addgene plasmid # 153083; http://n2t.net/addgene:153083; RRID:Addgene_153083). To generate plasmid pSL845, the $P_{y37a1b.5}$::mKate2 fragment amplified from pSL843 (primers: ods2785 and ods2787) and the *tram-1* fragment amplified from the *C. elegans* genomic DNA (primers: ods2788 and ods2789) were inserted into plasmid pLF3FShC through Sap I Golden Gate Assembly (*Nonet, 2020*).

Primers sequences are:

ods2785: 5'-CCGTAAGCTCTTCGTGGGTTGCAGAAAAATATTTCACTGTTTC-3';
ods2787: 5'-GCAACAGCTCTTCGCTCCGGAACCTCCACGGTGTCCGAGCTTGGA-3';
ods2788 5'-GCAACAGCTCTTCGGAGGTGGATCTGGAGGTGTTAAGCCGCAAGGAGG-3';
ods2789: 5'-GCAACAGCTCTTCGTACATGTAATAAAATATAAGAAAACGCTG-3'.

## CRISPR/Cas9 mediated gene editing

Knock-in strains were generated using a *dpy-10* Co-CRISPR strategy (*Arribere et al., 2014*; *Paix et al., 2015*; *Paix et al., 2017*). All crRNA and repair template sequences are in *Table 2*. An injection mix containing 0.12 µl *dpy-10* crRNA (0.6 mM) (Horizon Discovery/Dharmacon), 0.3 µl target gene crRNA (0.6 mM) for one locus editing or 0.21 µl of each crRNA (0.6 mM) for multi-loci editing and 1.46 µl (one locus) or 1.88 µl (two loci) universal tracrRNA (0.17 mM) (Horizon Discovery/Dharmacon) precomplexed with purified 7.6 µl of 40 µM Cas9 protein (UC Berkeley QB3) and 0.29 µl of the *dpy-10* single-strand DNA oligonucleotide (ssODN) (500 ng/µl) repair templates and 0.21 µl ssODN repair template (25 µM) for the target gene editing or up to 500 ng double-strand DNA were injected to the germline of the hermaphrodite young adults. For *anc-1(yc52[Δ25KASH])* I, *anc-1 (yc69[Δ29KASH])* I, *anc-1(yc62[ΔF1])* I, *anc-1(yc61[Δ6RPS])* I, *anc-1(yc80[ΔF2])* I, *anc-1(yc78[Δ5RPS])* I, *anc-1(yc91[gfp::anc-1b::Δ5rps])* I, *anc-1(yc70[gfp::anc-1b::Δ6rps])* I, *anc-1(yc92[gfp::anc-1b::Δf1])* I, *anc-1(yc93[anc-1AI972,973**::gfp])* I, *anc-1(yc106[gfp::anc-1b::Δkash])* I, *anc-1(yc94[gfp::anc-1b::Δ tk])*, I single-strand DNA (SSD) (synthesized by Integrated DNA Technologies, IDT) was used as repair template. For GFP knock-in strains, double-strand DNA repair templates were amplified with PCR from the plasmids pSL779 for *gfp* using Phusion polymerase and the primers listed in *Table 2* (New England Biolabs) (*Bone et al., 2016*; *Cain et al., 2018*).

Strains *anc-1[yc41(anc-1::gfp3Xflg::kash,Δch)]I* and *anc-1[yc36(anc-1::gfp3Xflag::kash)]I* were generated by Dickinson Self-Excising Drug Selection Cassette (SEC) method (*Dickinson et al., 2015*). In *anc-1[yc41(anc-1::gfp3Xflg::kash,Δch)]I*, both CH domains were deleted (starting with 23KAQK26 and ending with 322QFVR325) and replaced with GFP flanked with 9-residue long linkers (GASGASGAS).

## Microscopy and imaging analysis

Images of the nuclear anchorage, worm body size measurements and *anc-1b* promoter reporter assays were collected with a wide-field epifluorescent Leica DM6000 microscope with a 63 × Plan Apo 1.40 NA objective, a Leica DC350 FX camera, and Leica LAS AF software. ANC-1 subcellular localization, the ER marker, and nuclear shape images were taken with a spinning disc confocal microscope (Intelligent Imaging Innovations) with a CSU-X1 scan head (Yokogawa), a Cascade Quan-tEM 512SC camera (Photometrics), a 100 × NA 1.46 objective (Zeiss or Nikon), and SlideBook software (Intelligent Imaging Innovations). The contrast and levels of the images were uniformly adjusted using ImageJ (National Institutes of Health). Live GFP::KDEL images were acquired at 200 ms or 250 ms intervals using the above spinning disc confocal system. To quantify ER, lipid droplet, and mitochondria positioning defects, images from at least 10 young adults of each strain were scored blindly by three people. In addition, the 'Manual Tracking' plug-in for ImageJ (https://imagej.nih.gov/ij/plugins/track/track.html) was used to track the positions of multiple ER fragments and lipid droplets through a time-lapse series. The relative movements between three different spots per animal were measured over time.

To quantify the fluorescent intensity of GFP::ANC-1B and GFP::ANC-1(Δ6RPs), images were taken under the spinning disc confocal system (100X objective) and the RawIntDen of the was hypodermal area measured by ImageJ. The average fluorescent intensity was calculated by dividing the RawInt-Den by the area.

For the ER colocalization analysis, z stack images were taken under the spinning disc confocal system (100X objective) and processed by background subtraction with rolling ball radius of 50.0 pixels. A Region of interest (ROI) of 96 × 96 pixels in the non-nuclear region was cropped to use ScatterJ plugin for co-localization analysis (*Zeitvogel et al., 2016*).

For some adult animals, Image J 'Stitching' Plug-in was used to stitch images with overlap (*Preibisch et al., 2009*). The length of L2 larvae, width and length of the adult animals, as well as the circularity and solidity of the nuclei were measured with Image J.

**Table 2.** crRNA and repair templates used in this study.

| New alleles | Strain | crRNA * | DNA repair template * [†,§,‡] |
|---|---|---|---|
| anc-1(yc52[Δ25KASH]) I | N2 | CAGUACUCGUCG UCGCAAUG | GCACTGCTTGTTCTACTTATGGGAGCCGCTTGTTTGGTTCCACA cTGtGAtGAtGAaTAtTAATCTTTAATTTTTTATTTTTCATTACTATT CACTATTGTTTCATTCATCATGAACCTG |
| anc-1(yc69[Δ29KASH]) I | N2 | CAGUACUCGUCG UCGCAAUG | NHEJ [‡] |
| anc-1(yc41[anc-1::gfp3Xflag:: kash,Δch]) I | N2 | UUUCAUC UUGAAGAGGUUCG | –AAAATCTATTTTGAAAATTTTCAGATGAGGACGAG<EGFP-3xFLAG > AGATCAGGAGCTAGCGGAGCCATGTTCGGAGAAAGATCACCAATG– |
| anc-1(yc62[ΔF1]) I | N2 | ACUUGAUCAAUCUA UAAUAA | GAAACATGAAAGCAAAGTACATTTTTTTAAAAATCGATT ATTTCagATcGAcCAgGTACAGTCTGAGATCGACACTCTTT CAGACTTCGAGGAGATCGAGCGTGAAATAAACGGCTC ACTCGAAGCTTTCGAAGCCGAG |
| anc-1(yc61[Δ6RPS]) I | N2 | GCGUUCAAUUUC UUCAAAAU UGUUAGUA UUGGCGGCGAGU GGAGCGUUUUG UAAAAGCAA | AAGGTACAAAACATTGGAAAAACATCGATTGACGACG TGAATGTATCTGACTTCGAGGAGATCGAGCGTGAGAT CAATGGCTCCCTTGAGGCTTTCTCTATTTGGGAACG CTTCGTCAAGGCTAAAGATGATTTGTACGATTATTTG GAGAAATTAGAGAACAATGTAAGC |
| anc-1(yc80[ΔF2]) I | N2 | GGAGCGUUUUG UAAAAGCAA UCCAACGGGAUC UUUGUCGU | ACTCTTATTCCGGACCTTGAAGAAAGAGCTTCTATTTG GGAGCGTACTGCTTTGCCACTTCAGGTTTGTT TATATTTTTTAATATTAATA |
| anc-1(yc36[anc-1::gfp3Xflag:: kash]) I | N2 | GACAAAGATCCCG TTGGAGA | TCCGAC<u>GACAgAGATCtCGcTGGcGc</u>CGcGTACTCA GAACTTCAGGAGCTAGCGGAGCC < EGFP-3xFLAG > tcaggagctagcggagccGCTTTGCCACTTCAGgtttgtttatattttt |
| anc-1([yc68[gfp::anc-1b]]) I | N2 | CCGUCGGAACAGC UCCAUUU | TCTTAACCTTTTGTTCCATTCACTAATTATTTTCAAT TACAGGAGGTTGGCCGCGAGTCGGTCAGTAAATTA TCAGCTGCAGTTGACGATCGATACATCTACACGTT ACACGTGCTCCGAACTAAG < EGFP > GGAGGTTCCG GAGGTGGATCTGGAGGTGAaCTcTTtCGtCGtCTGCAA AACTTTTGCGACGCTGTCAAAATATTGCGATCGCAAAA TACCAAATGGAACGGAATCAAGATTTCGCAGGTTTGT TTCAAAAGCATCACAAATTAGCGG |
| anc-1(yc78[Δ5RPS]) I | N2 | GGAGCGUUUUG UAAAAGCAA UUCCUCUGGCUU- CAACGAGU | GATCAGCTCAAGTCGGACGATTTGAAGACGGCAGAA AAGGAAATCACTAAtagccTcAAaCCcGAaTCTATTTGG GAaaGaTTcGTtAAgGCtAAAGATGATTTGTACGATT ATTTGGAGAAATTAGAGAACAAT |
| anc-1(yc90[anc-1::gfp::F2]) I | N2 | GGAGCGUUUUG UAAAAGCAA | CCGGACCTTGAAGAAAGAGCTTCTATTTGGGAGCG TGGTGGAAGTGGTGGAGGAAGCGGTGGA < EGFP > GCATGGATGAACTATACAAAGGAGGTTCCGGAGGTGG ATCTGGAGG<u>TTTcGTcAAgGCt</u>AAAGATGATTTGTACG ATTATTTGGAGAAATTAGAGAACA |
| anc-1(yc70[gfp::anc-1b::Δ6rps]) I | UD612 | GCGUUCAAUUUC UUCAAAAU UGUUAGUA UUGGCGGCGAGU GGAGCGUUUUG UAAAAGCAA | AAGGTACAAAACATTGGAAAAACATCGATTGACGAC GTGAATGTATCTGACTTCGAGGAGATCGAGCGTGA GATCAATGGCTCCCTTGAGGCTTTCTCTATTTGGGA ACGCTTcGTcAAgGCtAAAGATGATTTGTACGATTAT TTGGAGAAATTAGAGAACAATGTAAGC |
| anc-1(yc91[gfp::anc-1b::Δ5rps]) I | UD612 | GGAGCGUUUUG UAAAAGCAA UUCCUCUGGCUU- CAACGAGU | GATCAGCTCAAGTCGGACGATTTGAAGACGGCAGA AAAGGAAATCACTAAtagccTcAAaCCcGAaTCTATTT GGGAaaGaTTcGTtAAgGCtAAAGATGATTTGTAC GATTATTTGGAGAAATTAGAGAACAAT |
| anc-1(yc92[gfp::anc-1b::Δf1]) I | UD612 | ACUUGAUCAAUCUA UAAUAA | GAAACATGAAAGCAAAGTACATTTTTTTAAAAATCG ATTATTTCagATcGAcCAgGTACAGTCTGAGATCGAC ACTCTTTCAGACTTCGAGGAGATCGAGCGTGAAAT AAACGGCTCACTCGAAGCTTTCGAAGCCGAG |
| anc-1(yc93[anc-1AI972,973**:: gfp]) I | UD612 | ACUCACCUCUAGAAA UUCGA | CAAAATTTAGAGCTCAGCAATGAGCAGGACTGTCC AGATtaatgaGgtaccCTAGAGGTGAGTATAGTCA TTTTCCGCTCATTACACTCTT |
| anc-1(yc71[ΔTK]) I | N2 | CAGAACUGC UUUGCCACUUC AUUAAAGAUUAAAA UGGUGG | GAACAACTCCGACGACAAAGATCCCGTTGGAGACG GGTACTCAGATAATCTTTAATTTTTTATTTTCATT ACTATTCACTATTGTTTCATTCATC |

*Table 2 continued on next page*

*Table 2 continued*

| New alleles | Strain | crRNA * | DNA repair template * [†,§,‡] |
|---|---|---|---|
| anc-1(yc106[gfp::anc-1b::Δkash]) I | UD612 | CAGUACUCGUCGUCGCAAUG | GCACTGCTTGTTCTACTTATGGGAGCCGCTTGTTTGGTTCCACAcTGtGAtGAtGAaTAtTAATCTTTAATTTTTTATTTTCATTACTATTCACTATTGTTTCATTCATCATGAACCTG |
| anc-1(yc94[gfp::anc-1b::Δtk]) | UD612 | CAGAACUGCUUUGCCACUUCAUUAAAGAUUAAAAUGGUGG | GAACAACTCCGACGACAAAGATCCCGTTGGAGACGGGTACTCAGATAATCTTTAATTTTTTATTTTCATTACTATTCACTATTGTTTCATTCATC |
| anc-1(yc73[gfp::anc-1b::Δ6rps::Δtk]) I | UD618 | CAGAACUGCUUUGCCACUUCAUUAAAGAUUAAAAUGGUGG | GAACAACTCCGACGACAAAGATCCCGTTGGAGACGGGTACTCAGATAATCTTTAATTTTTTATTTTCATTACTATTCACTATTGTTTCATTCATC |
| dpy-10(cn64) | Co-CRISPR | GCUACCAUAGGCACCACGAG *Arribere et al., 2014* | CACTTGAACTTCAATACGGCAAGATGAGAATGACTGGAAACCGTACCGCATGCGGTGCCTATGGTAGCGGAGCTTCACATGGCTTCAGACCAACAGCCTAT (*Arribere et al., 2014*) |

*all nucleotide sequences are displayed as single strand in the 5' to 3' orientation.

†In many cases a ssDNA oligonucleotide was used. For larger inserts, a PCR product was used.

§An imprecise NHEJ event led to an in-frame deletion without using the repair template.

¶Underlined sequences introduce silent mutations so the repair template is not cut by Cas9.

‡Underline indicates the silent mutation in the repair template.

## Statistical evaluation

The nuclear anchorage quantifying data were displayed as scatter plots with means and 95% CI as error bars. Sample sizes are indicated in the figures. The statistical tests are indicated in the figure legends. When there were limited comparisons. unpaired student t-tests were performed on the indicated comparisons for the nuclear anchorage assay, and Fisher's exact test was used. When multiple comparisons were made, ANOVA and Tukey's multiple comparisons tests were used. Prism nine software was used for the statistical analyses.

## Acknowledgements

We thank Gant Luxton, Erin Cram, Charlotte Kelley, and members of the Starr lab, for helpful discussions and editing of the paper. We thank Barth Grant for sharing an unpublished strain, Erin Tapley and Yu-Tai Chang for the mitochondrial GFP strain, and Joshua Morgan for suggestions on nuclear shape measurements. We thank Michael Paddy at the MCB Light Microscopy Imaging Facility, which is a UC Davis Campus Core Research Facility, for microscopy assistance. The 3i Marianas spinning disc confocal used in this study was purchased using NIH Shared Instrumentation Grant 1S10RR024543-01. These studies were supported by the National Institutes of Health grants R01GM073874 and R35GM134859 to DAS.

## Additional information

### Funding

| Funder | Grant reference number | Author |
|---|---|---|
| National Institutes of Health | R01GM073874 | Daniel A Starr |
| National Institutes of Health | R35GM134859 | Daniel A Starr |

The funders had no role in study design, data collection and interpretation, or the decision to submit the work for publication.

## Author contributions
Hongyan Hao, Conceptualization, Resources, Data curation, Formal analysis, Supervision, Validation, Investigation, Visualization, Methodology, Writing - original draft, Project administration, Writing - review and editing; Shilpi Kalra, Conceptualization, Formal analysis, Validation, Investigation, Methodology, Writing - review and editing; Laura E Jameson, Resources, Data curation, Formal analysis, Investigation, Writing - review and editing; Leslie A Guerrero, Natalie E Cain, Conceptualization, Resources, Formal analysis, Investigation, Methodology, Writing - review and editing; Jessica Bolivar, Conceptualization, Resources, Investigation, Methodology; Daniel A Starr, Conceptualization, Data curation, Formal analysis, Supervision, Funding acquisition, Validation, Visualization, Methodology, Writing - original draft, Project administration, Writing - review and editing

## Author ORCIDs
Hongyan Hao (iD) https://orcid.org/0000-0003-0860-2615
Natalie E Cain (iD) http://orcid.org/0000-0003-1391-404X
Daniel A Starr (iD) https://orcid.org/0000-0001-7339-6606

## Decision letter and Author response
Decision letter https://doi.org/10.7554/eLife.61069.sa1
Author response https://doi.org/10.7554/eLife.61069.sa2

# Additional files

## Supplementary files
• Transparent reporting form

## Data availability
The list of strains generated is detailed in Table 1. All data points are displayed in the histograms in the figures.

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
