## [Decision Letter]

**Acceptance summary:**

The reviewers all felt your study represents a valuable contribution to enable better understanding of the organization of organelles in the cell and to encourage a re-think of the current models of nuclear anchorage.

**Decision letter after peer review:**

Thank you for submitting your article "The Nesprin-1/-2 ortholog ANC-1 regulates organelle positioning in *C. elegans* without its KASH or actin-binding domains" for consideration by *eLife*. Your article has been reviewed by 3 peer reviewers, one of whom is a member of our Board of Reviewing Editors, and the evaluation has been overseen by Piali Sengupta as the Senior Editor. The following individual involved in review of your submission has agreed to reveal their identity: Verena Jantsch (Reviewer #2).

The reviewers have discussed the reviews with one another and the Reviewing Editor has drafted this decision to help you prepare a revised submission.

Summary:

The current and widely accepted model how nuclei are positioned in the cell proposes that their anchorage is mediated through nuclear membrane spanning SUN and KASH domain containing proteins which connect the cytoskeleton to the nuclear interior. This study presents evidence that this model needs reconsidering. The authors follow up on their previous observation that nuclear positioning differs between mutants in SUN (UNC-84) and KASH proteins (ANC-1), where anc-1 mutants display a more severe phenotype. The study nicely combines structure-function analysis of ANC-1 with the readout of nuclei positioned in the syncytial cell hyp7 in fixed samples and by live imaging, and the mutant analysis correlates well with the fitness of the whole organism. The authors also show that the expected regions (the KASH domain or the actin binding domain) play rather minor roles in nuclear anchorage. Instead, anchorage is mapped to a portion of the transmembrane domain and to spectrin-like repeats in the cytoplasmic localised region of the protein. Mutating those protein domains not only leads to irregularly positioned nuclei but also to disorganisation and fragmentation of the ER, which also appears unanchored as evidenced by in vivo imaging. Consistently, the authors find ANC-1 prominently associated with the ER and that this association is independent of the KASH domain.

This paper will be very interesting to a wide readership as it challenges current models of nuclear positioning and shows that mis-positioning and loss of connectivity can have severe consequences to the development of the organism. However, the reviewers felt that some of the conclusions required additional data to support the statements made, in particular to strengthen the proposed the ER link, as well as some other points that require clarification as detailed below:

Essential revisions:

1. The data shown in Fig6 suggesting ANC-1 targets to the ER was not convincing and was also very hard to contextualise as no other markers for other organelles or the nucleus were present, and no merged images provided. Given that the authors also repeatedly draw conclusions relating to mitochondria, including a mitotracker in the far-red channel would be important. The ANC1 signal appeared rather diffusely localised in all cases (WT and TK/KASH mutants; noting that the KDEL signal is very low in the TK mutant image compared to the others) and it could be argued any potential overlap might be coincidental rather than specific. No other evidence for ER targeting was provided but would be required to support the current conclusions. The authors should also at minimum calculate the co-localization index and provide examples of line scans to show a correlation between signal intensities. Also the sentence "The slight discrepancy between localization of mKate2::ANC-1b and GFP::KDEL could be explained by the distance between the ANC-1 N terminus and the ER membrane" will need to be modified. This is highly unlikely due to the resolution of the microscope used to acquire this image. The discrepancy might be due to differences in the focus plane of the two light channels. Please remove or edit this speculation.

2. A further concern is the lack of any cytoskeletal network analysis in the paper; the authors (somewhat extrapolated) conclusions are focused on the concept that nuclei and the ER are detached from any sort of cytoplasmic structural network in anc-1 mutants, but this is not tested in the study. It would be very helpful to provide this evidence, but we recognise that analysing the cytoskeleton in detail may prove impractical given the current restrictions. However, if experimental data cannot be provided, it would be important to tone down such statements given the lack of formal evidence provided for this model.

3. ANC1b-Δ6RPS and Δ5RPs appear to be expressed at much lower levels than other mutants (or WT protein) eg: Figure 5; is this the case and if so, might the nuclear positioning defect just be due to this (ie: as the animals are more like anc-1 null)? It would be important to understand relative stability of these mutant proteins to interpret this data. Please provide more information on the relative intensity levels of each mutant and if they are similar, provide more representative example images.

4. The authors put forward the claim that ANC-1 anchors other organelles besides the ER. There are a lot of markers available for different organelles, and it would be important for the authors to provide some staining in the most significant mutants (Δ6rps, Δtk), for mitochondria or other organelles. On page 11 the authors also write "Nuclear shape changes were observed during live imaging in anc-1 mutants consistent with a model where anc-1 mutant nuclei are susceptible to pressures from the cytoplasm, perhaps crashing into lipid droplets that corresponded with dents in nuclei.". It is not clear why the authors suggest "crashing into lipid droplets" and not any other organelles in the cell (mitochondria, endosomes, etc). Is there any evidence for this statement that can be provided?

5. p. 14:.…was not enriched at the nuclear envelope…". it would very important to include pictures of the ANC-1 staining together with a marker for the nuclear envelope in Figure 5 to support this statement.

6. Page 6, figure 2A,B. it is not clear in the text that two of the four nonsense mutations that were analysed only disrupt isoform a and c (w427 and w621). If these mutations only affect isoform a and c, and not the shorter isoform b, then the authors could explain this better, so that this result goes together with the RNAi data.

7. Figure 1c and Figure 3C – the dataset of WT and anc-1(e1873) seems to be the same in both these figures. Although duplication of the same data in multiple figures should not typically occur in publications, the duplication should be indicated clearly in the figure legend for transparency. Ideally, the duplicated data should also be shown in a different color/format in the figure so that they are immediately obvious. Other duplications of data in the manuscript should also be indicated. Why was unc-84(n369) data repeated three times on the graph in Figure 3C but with different values?

8. The percentage of touching nuclei in unc84(n369), KASH mutants (Figure 1C), and anc-1(dF1)(Figure 3c), are around 20%. However, the authors classify the KASH mutants as "mild nuclear anchorage", whereas anc-1(dF1) on Figure 3c was classified as "severe nuclear anchorage". The authors also used the term "intermediate nuclear anchorage" for other phenotypes. The authors should use the same classification (clearly described) throughout the manuscript.

9. Figure 4A – the authors described the image of GFP:Kdel of hyp7 syncytia as a "branched network". It is very difficult to observe any branches and a network in these images. We recognise the authors want to use the same terminology that is used to characterize ER in other cells, but it is difficult to understand what the authors mean by a branch. Diffuse staining with dark round objects in it is visible, but branches are not clear. The authors should provide a better example, more similar to unc-84(n369) mutant, where branches and a network are more distinct and annotate these properly to support this statement.

10. Figure 4C and D; Video 1 and 2- In these figures and the authors compare the dynamics of ER in WT and anc-1 mutant. The authors also provide representative videos. The WT animal appears to be crawling significantly less than the anc-1 mutant, making it difficult to directly compare the anchorage of ER in WT during crawling in the representative videos. Moreover, – Supp Video 1 appears to be far shorter (fewer frames) than Supp Video 2 – ideally, equivalent time scales should be provided for WT animals to be comparable with the mutant.

11. Figure 6A- in this scheme and through the manuscript, the authors refer to the c-term region of ANC-1, after the TM domain, as the KASH domain. However, by definition, the KASH domain includes part of the neck region, the TM domain and the c-term region after the TM domain (see e.g. Figure 4 of Wihelmsen et al. JCS 2006, or PFAM definition of KASH – 10541). Therefore, the neck region of the KASH domain plays a role in nuclear positioning. Furthermore, labelling of figure 6A should be changed accordingly to standard definition.

---

## [Author Response]

Essential revisions:1. The data shown in Fig6 suggesting ANC-1 targets to the ER was not convincing and was also very hard to contextualise as no other markers for other organelles or the nucleus were present, and no merged images provided. Given that the authors also repeatedly draw conclusions relating to mitochondria, including a mitotracker in the far-red channel would be important. The ANC1 signal appeared rather diffusely localised in all cases (WT and TK/KASH mutants; noting that the KDEL signal is very low in the TK mutant image compared to the others) and it could be argued any potential overlap might be coincidental rather than specific. No other evidence for ER targeting was provided but would be required to support the current conclusions. The authors should also at minimum calculate the co-localization index and provide examples of line scans to show a correlation between signal intensities. Also the sentence "The slight discrepancy between localization of mKate2::ANC-1b and GFP::KDEL could be explained by the distance between the ANC-1 N terminus and the ER membrane" will need to be modified. This is highly unlikely due to the resolution of the microscope used to acquire this image. The discrepancy might be due to differences in the focus plane of the two light channels. Please remove or edit this speculation.

We have substantially improved our description of the co-localization of ANC-1 with organelles. We removed the old Figure 6D-F and present an entirely new Figure 7 showing the co-localization of ANC-1 is consistent with (but not entirely overlapping) the ER. In the old analysis, the GFP marker of the ER lumen was overexpressed and significantly more bright than the weakly expressing mKate::ANC-1. We therefore generated a new single-copy hypodermal-specific mKate2::TRAM-1 ER membrane marker strain and crossed the marker into our GFP::ANC-1 wild-type and mutant strains. These two markers were expressed at closer to equal levels, making co-localization easier. Thus, using entirely different lines and markers, we obtained similar results, increasing the rigor and reproducibility of our findings. GFP-ANC-1b localized similar to, but not exactly overlapping with, the ER. We used ImageJ plug-in ScatterJ to quantify the co-localization and the Pearson's correlation coefficient is shown. We now also include merged images. These data are now discussed on pages 12.

We also examined the localization of GFP::ANC-1b with relation to mitochondria and lipid droplets (Figure 7—figure supplement 1). ANC-1 clearly localizes in a pattern very different from mitochondria or lipid droplets. See text on page 12.

As suggested, the sentence "The slight discrepancy between localization of mKate2::ANC-1b and GFP::KDEL could be explained by the distance between the ANC-1 N terminus and the ER membrane" has been removed. We agree that it was too speculative.

2. A further concern is the lack of any cytoskeletal network analysis in the paper; the authors (somewhat extrapolated) conclusions are focused on the concept that nuclei and the ER are detached from any sort of cytoplasmic structural network in anc-1 mutants, but this is not tested in the study. It would be very helpful to provide this evidence, but we recognise that analysing the cytoskeleton in detail may prove impractical given the current restrictions. However, if experimental data cannot be provided, it would be important to tone down such statements given the lack of formal evidence provided for this model.

The *C. elegans* adult hypodermal actin cytoskeleton is hard to visualize using current tools. For example, to see how disorganized the actin cytoskeleton is in wild type adult hypodermis, see Figures 1E and 2C of Higuchi-Sanabria, Ryo, et al. "Spatial regulation of the actin cytoskeleton by HSF-1 during aging." Molecular biology of the cell 29.21 (2018): 2522-2527. We decided that new tools need to be created to study the actin cytoskeleton in the hypodermis that would be beyond the scope of this manuscript.

We obtained a GFP::MAPH-1.1 line from Mike Boxem’s lab that we used to visualize microtubules in the adult hypodermis. As previously described, in wildtype, GFP::MAPH-1.1 localizes to non-centrosomal microtubules in the hypodermis. The density and disorganization of the microtubule network in wildtype makes it very difficult to quantify any parameters. Nonetheless, we can qualitatively state that in *anc-1* mutant animals, most of the hyp7 syncytium the microtubules appear normal. The exception is where nuclei have moved back and forth along an anterior-posterior axis, they appear to have cleared a microtubule free channel. We suspect this is a result of untethered nuclei plowing through and is more likely a result of unanchored nuclear movements than the cause of nuclear mislocalization. We now include three new Video (8-10) and a new Figure 10 with still images showing hypodermal microtubule organization in wild type and *anc-1* mutants. Text on pages 13-14 describes these data.

We also attempted to tone down the role of the cytoskeleton in our fairly speculative model in the last paragraph of the Discussion. Clearly, this study raises many new questions that will require extensive future studies.

3. ANC1b-Δ6RPS and Δ5RPs appear to be expressed at much lower levels than other mutants (or WT protein) eg: Figure 5; is this the case and if so, might the nuclear positioning defect just be due to this (ie: as the animals are more like anc-1 null)? It would be important to understand relative stability of these mutant proteins to interpret this data. Please provide more information on the relative intensity levels of each mutant and if they are similar, provide more representative example images.

We took new images under the exact same conditions and carefully measured the total fluorescence of the wild-type GFP::ANC-1b and the Δ6RPs. As suggested by our original images, there is significantly less expression of the mutant ANC-1 protein. We show the data in the new Supplemental Figure 3. We add the following sentences on the top of page 10 to discuss the levels: “However, the intensity of the six repeat deletion mutant is significantly less than wild-type GFP::ANC-1b (Figure 5—figure supplement 1), making it possible that the phenotype is due to less ANC-1 being present. Yet, the deletion mutant is significantly enriched at the nuclear envelope (Figure 5—figure supplement 1), supporting the hypothesis the defect is due to a loss of ANC-1 repeats at the general ER.”

4. The authors put forward the claim that ANC-1 anchors other organelles besides the ER. There are a lot of markers available for different organelles, and it would be important for the authors to provide some staining in the most significant mutants (Δ6rps, Δtk), for mitochondria or other organelles. On page 11 the authors also write "Nuclear shape changes were observed during live imaging in anc-1 mutants consistent with a model where anc-1 mutant nuclei are susceptible to pressures from the cytoplasm, perhaps crashing into lipid droplets that corresponded with dents in nuclei.". It is not clear why the authors suggest "crashing into lipid droplets" and not any other organelles in the cell (mitochondria, endosomes, etc). Is there any evidence for this statement that can be provided?

Thank you for this suggestion. We now show in new figures 8 and 9 and in Videos 4-7 that the mitochondria and lipid droplets have similar positioning defects as the ER. These data significantly increase the impact of the manuscript and are discussed on page 13.

We have toned down the speculative parts of the model on page 13 and deleted the mention of the possibility that lipid droplets are the sole cause of nuclear envelope dents.

5. p. 14:.…was not enriched at the nuclear envelope…". it would very important to include pictures of the ANC-1 staining together with a marker for the nuclear envelope in Figure 5 to support this statement.

We have added to data to the Figure 5—figure supplement 1 showing a merge of wild-type or mutant GFP::ANC-1b with a nuclear envelope marker. These data are mentioned in the results on pages 10 and referenced in the discussion on the top of page 17 in the sentence referenced by the reviewers for this point.

6. Page 6, figure 2A,B. it is not clear in the text that two of the four nonsense mutations that were analysed only disrupt isoform a and c (w427 and w621). If these mutations only affect isoform a and c, and not the shorter isoform b, then the authors could explain this better, so that this result goes together with the RNAi data.

We clarified the text pertaining to these two nonsense mutants on page 6.

7. Figure 1c and Figure 3C – the dataset of WT and anc-1(e1873) seems to be the same in both these figures. Although duplication of the same data in multiple figures should not typically occur in publications, the duplication should be indicated clearly in the figure legend for transparency. Ideally, the duplicated data should also be shown in a different color/format in the figure so that they are immediately obvious. Other duplications of data in the manuscript should also be indicated. Why was unc-84(n369) data repeated three times on the graph in Figure 3C but with different values?

Thank you for pointing this out. Some data (the first three columns) in Figure 1C were repeated in Figure 3C for easy reference. We have made this clear in the figure legend for Figure 3C by adding the following sentence: “The data in the first three columns, *WT*, *anc-1(e1873)*, and *unc-84(n369)* are duplicated from Figure 1C and copied here for easy reference.” In addition, some of these data are copied in Figure 6B. This is now clearly stated in the Figure 6 legend.

The other two entries in Figure 3C for *unc-84(n369)* lost the labels for the second mutation that is also in those strains. We have corrected the labeling in 3C. Thanks for catching this mistake that made the figure difficult to comprehend.

8. The percentage of touching nuclei in unc84(n369), KASH mutants (Figure 1C), and anc-1(dF1)(Figure 3c), are around 20%. However, the authors classify the KASH mutants as "mild nuclear anchorage", whereas anc-1(dF1) on Figure 3c was classified as "severe nuclear anchorage". The authors also used the term "intermediate nuclear anchorage" for other phenotypes. The authors should use the same classification (clearly described) throughout the manuscript.

Thank you for this suggestion, which greatly clarifies the manuscript. On page 5 we add the following sentence: “Throughout this manuscript, we call nuclear anchorage defects that are statistically similar to *anc-1* null mutants as severe, defects statistically similar to *unc-84* null mutants but still significantly worse than wildtype as mild, and defects as statistically between *anc-1* and *unc-84* null mutants as intermediate.” We are now consistent in our use of severe, mild and intermediate.

9. Figure 4A – the authors described the image of GFP:Kdel of hyp7 syncytia as a "branched network". It is very difficult to observe any branches and a network in these images. We recognise the authors want to use the same terminology that is used to characterize ER in other cells, but it is difficult to understand what the authors mean by a branch. Diffuse staining with dark round objects in it is visible, but branches are not clear. The authors should provide a better example, more similar to unc-84(n369) mutant, where branches and a network are more distinct and annotate these properly to support this statement.

We have changed our description of the morphology of the ER on page 8. We no longer characterize the ER network as branched in wildtype. It’s simply the normal ER network. We are not trying to describe the ER in hypodermal cells. Our main point is that the ER is disrupted in the mutants. We therefore now avoid the term branched.

10. Figure 4C and D; Video 1 and 2- In these figures and the authors compare the dynamics of ER in WT and anc-1 mutant. The authors also provide representative vidoes. The WT animal appears to be crawling significantly less than the anc-1 mutant, making it difficult to directly compare the anchorage of ER in WT during crawling in the representative videos. Moreover, – Supp Video 1 appears to be far shorter (fewer frames) than Supp Video 2 – ideally, equivalent time scales should be provided for WT animals to be comparable with the mutant.

We have changed Video 1 for a wild-type example better matched with the amount of worm crawling is seem in Video 2.

11. Figure 6A- in this scheme and through the manuscript, the authors refer to the c-term region of ANC-1, after the TM domain, as the KASH domain. However, by definition, the KASH domain includes part of the neck region, the TM domain and the c-term region after the TM domain (see e.g. Figure 4 of Wihelmsen et al. JCS 2006, or PFAM definition of KASH – 10541). Therefore, the neck region of the KASH domain plays a role in nuclear positioning. Furthermore, labelling of figure 6A should be changed accordingly to standard definition.

We changed the labeling of Figure 6A to focus on the “luminal domain.” When we are specifically talking about the luminal half of KASH but not the trans-membrane span, we are much clearer now, using the word luminal to describe the deletions throughout the manuscript.